# The evolution of the type VI secretion system as a disintegration weapon

William P. J. Smith[1,2☯], Andrea Vettiger[3☯], Julius Winter[3,4], Till Ryser[3], Laurie E. Comstock[5], Marek Basler[3]*, Kevin R. Foster[1,2]*

**1** Department of Biochemistry, University of Oxford, Oxford, United Kingdom, **2** Department of Zoology, University of Oxford, Oxford, United Kingdom, **3** Biozentrum, University of Basel, Basel, Switzerland, **4** École Polytechnique Fédérale de Lausanne, Lausanne, Switzerland, **5** Brigham and Women's Hospital, Harvard Medical School, Boston, Massachusetts, United States of America

☯ These authors contributed equally to this work.
\* marek.basler@unibas.ch (MB); kevin.foster@zoo.ox.ac.uk (KRF)

**Data Availability Statement:** Our raw data are available to download from the FigShare file repository (dx.doi.org/10.6084/m9.figshare.11980491).

## Abstract

The type VI secretion system (T6SS) is a nanomachine used by many bacteria to drive a toxin-laden needle into other bacterial cells. Although the potential to influence bacterial competition is clear, the fitness impacts of wielding a T6SS are not well understood. Here we present a new agent-based model that enables detailed study of the evolutionary costs and benefits of T6SS weaponry during competition with other bacteria. Our model identifies a key problem with the T6SS. Because of its short range, T6SS activity becomes self-limiting, as dead cells accumulate in its way, forming "corpse barriers" that block further attacks. However, further exploration with the model presented a solution to this problem: if injected toxins can quickly lyse target cells in addition to killing them, the T6SS becomes a much more effective weapon. We tested this prediction with single-cell analysis of combat between T6SS-wielding *Acinetobacter baylyi* and T6SS-sensitive *Escherichia coli*. As predicted, delivery of lytic toxins is highly effective, whereas nonlytic toxins leave large patches of *E. coli* alive. We then analyzed hundreds of bacterial species using published genomic data, which suggest that the great majority of T6SS-wielding species do indeed use lytic toxins, indicative of a general principle underlying weapon evolution. Our work suggests that, in the T6SS, bacteria have evolved a disintegration weapon whose effectiveness often rests upon the ability to break up competitors. Understanding the evolutionary function of bacterial weapons can help in the design of probiotics that can both establish well and eliminate problem species.

## Introduction

Bacteria are aggressive organisms that employ a wide range of mechanisms to kill their competitors [1–3]. One of the most widespread and intricate of these anticompetitor adaptations is the type VI secretion system (T6SS) [4–6]. Resembling a spring-loaded spear gun, a T6SS is a contractile nanomachine that typically functions to inject toxic effector proteins into prokaryotic and eukaryotic cells. When a bacterial cell's T6SS is fired, a molecular spring drives a

**Funding:** WPJS, KRF, and LEC are supported by the National Institutes of Health (https://www.nih.gov/, project number 2R01AI093771-05). AV was supported by the Biozentrum Basel International PhD Program "Fellowships for Excellence"(https://www.biozentrum.unibas.ch/phd/international-phd-program/phd-program/). JW, TR, and MB are supported by an SNSF Starting Grant (http://www.snf.ch/, BSSGI0_155778/1). KRF is funded by the European Research Council (https://erc.europa.eu/, grant 787932) and a Wellcome Trust Investigator award (https://wellcome.ac.uk/, 209397/Z/17/Z). The funders had no role in study design, data collection and analysis, decision to publish, or preparation of the manuscript.

**Competing interests:** The authors have declared that no competing interests exist.

**Abbreviations:** CDI, contact-dependent growth inhibition; CFU, colony-forming unit; GPU, graphics processing unit; Hcp, hemolysin-coregulated protein; T6SS, type VI secretion system; Tae1, type VI amidase effector 1; Tse2, type VI effector 2.

poison-tipped needle outward, potentially piercing and intoxicating a nearby target cell [7,8]. Cells are spared damage from clonemates' T6SS attacks through the synthesis of shared immunity proteins, but cells lacking immunity are intoxicated. Though dependent on cell–cell contact, this delivery route typically confers a broad target spectrum, enabling toxin translocation without relying on target cells' surface receptors or transport systems [9,10].

Commensurate with these advantages, there is growing evidence of the important role the T6SS plays in microbial ecosystems. The T6SS can be a powerful mediator of interbacterial competition across a broad range of contexts, including plant- and human-associated communities [3,11–13]. In these settings, it facilitates pathogen invasion through the killing of commensal species—but, conversely, it can allow commensals to defend ecological niches from competitors and pathogens [14–17]. Understanding T6SS function and evolution, therefore, has implications for both crop protection and the treatment of infectious diseases.

Although the ecological impacts of T6SSs are becoming clear, the evolutionary pressures on the T6SS as an anticompetitor weapon remain poorly understood. Killing competitors offers fitness benefits from improved access to space or resources [2]—but, as with other weapons [2,18,19], there appear to be costs to T6SS use. Firing likely comes with high material and energetic overheads [20], and the scope for T6SS component recycling is limited [21]. The arrangement of different strains in space also affects the utility of the weapon, by changing the proportion of attacks directed against non-kin cells [22,23]. This prompts the question: How should the T6SS be used, if at all, during bacterial competition?

To explore this question, we developed a detailed agent-based model of T6SS competition. Our approach highlights a major limitation to the T6SS's contact-dependent mode of killing. Although increasing the rate of T6SS needle firing can benefit a cell through the elimination of competitors, killing rapidly becomes self-limiting, because dead cells accumulate to form protective barriers around groups of T6SS-sensitive cells. This led us to a key prediction on the design of the T6SS as a weapon: it will only remove competitors effectively if it delivers lytic toxins that not only kill but also disintegrate target cells. This prediction is supported by competition experiments, both in microfluidics and on agar, and by the distribution of lytic toxins across a wide range of T6SS-wielding bacterial species. Our work reveals that the T6SS suffers a major design constraint but that this can be overcome by the delivery of toxins with a particular mechanism of action. We discuss the implications of our findings both for T6SS evolution and for the goal of using probiotic bacteria to deliver antimicrobials.

## Results and discussion

### A new agent-based model of T6SS competition

When is the T6SS favored by natural selection as a mechanism to kill bacterial competitors? To address this question, we created a realistic agent-based model incorporating T6SS dynamics. We chose this approach as it allows one to freely alter both environmental and weapon properties—such as cell density, firing rate, the cost of firing, and toxin potency—while retaining many of the spatial and mechanical details that are known to be important for bacterial competition [23,24].

We use our model to explore where and how T6SS use affects bacterial fitness, across a wide range of conditions. As in previous studies [25–27], we consider rod-shaped bacterial cells that push on one another as they grow and divide exponentially (S1A Fig, "Dynamics"). Model simulations begin with cells randomly scattered on a flat surface ($t_{start}$) and end once the cell population reaches a user-defined maximum (at time $t_{end}$). To this established framework, we added a discrete representation of T6SS firing and response. T6SS+ "attacker" cells (green) fire needles from randomly chosen sites on their surface, at an average rate $k_{fire}$. For

simplicity, we assume that T6SS activity is constitutive (constant $k_{fire}$); this appears to be the case for some bacteria under certain conditions [28,29], although others are known to use sophisticated regulatory strategies to adjust firing rates in response to biotic [28,30,31] and abiotic [32] cues. Attacker cells also pay a fractional growth cost proportional to their firing rate, so that the average T6SS+ cell growth rate is $k_{grow}(1-ck_{fire})$. T6SS− susceptible (magenta) avoid these costs but become intoxicated after being struck by $N_{hits}$ needles. Intoxicated "victims" (black) cannot grow and are lysed (removed from the simulation) following a delay of $1/k_{lysis}$. Our model, whose source code is available to download [33], is based on CellModeller (Tim Rudge and colleagues, https://haselofflab.github.io/CellModeller/), a GPU-compatible Python/ OpenCL modeling framework [25,34,35]. Further details are provided in the Materials and methods section, and in S1 Fig.

## A constraint to T6SS use: The corpse barrier effect

Using this model, we simulated competitions between attacker and susceptible strains growing in a single layer. We initiated each simulation by scattering a 1:1 (100-cell) mixture of the 2 cell types in a circle of 50-μm radius, repeating simulations for increasing T6SS firing rates $k_{fire}$. Surprisingly, this revealed that high T6SS activity can cause an attacker strain to lose, such that the T6SS is counterselected (Fig 1A, S1 Fig, S2 Fig and S1 Movie). Analyzing our simulations in more detail, we noticed that as T6SS firing increases, so too did the number of dying victim cells found at the interstrain boundary (Fig 1B and S2 Fig). This finding is consistent with previous empirical observations of dead cells between mutually vulnerable T6SS+ attacker strains [36,37] and led us to hypothesize that lysing cells might block T6SS attacks at the interstrain boundary, thereby reducing the efficiency of the weapon. We refer to this process as the "corpse barrier effect".

Consistent with this hypothesis, we found that decreasing the duration of victim lysis (parameterized by $1/k_{lysis}$) increased the benefits of firing the T6SS rapidly (Fig 1C and 1G) and lowered boundary saturation, defined as the fraction of the interface occupied by lysing cells (Fig 1D and S2 Fig). We also found similar behavior in analogous 3D simulations (Fig 1E and 1F). As a further mechanistic test, we quantified the benefits of T6SS activity as a function of attacker cells' firing rate. We calculated the peak T6SS killing rate in each competition and normalized it by the number of cells on interstrain boundaries (see Materials and methods and S2 Fig). These values confirmed that killing rapidly saturates with $k_{fire}$ when victim lysis is slow (Fig 1H, blue markers) but is substantially increased when victim lysis is rapid (red markers) following intoxication. Importantly, we also found that we could accurately predict observed killing rates simply from boundary saturation values (see Materials and methods; Fig 1H, dashed lines), suggesting that "corpse barriers" are indeed a determinant of overall T6SS killing efficiency.

Taken alongside the rising costs of T6SS investment, these kill rate values provide a rationale for the observed fitness effects (Fig 1G). For a given inoculum density and composition, the saturating benefits of T6SS use are rapidly overtaken by rising weapon costs. This result suggested that the corpse barrier effect is a fundamental constraint on the T6SS as a weapon. However, these analyses also indicate a possible solution to this problem: decreasing the duration of victim cell lysis is predicted to greatly improve the rate of killing via the T6SS across a wide range of conditions (Fig 1G and 1H, and S2 Fig). This predicts that cells wielding the T6SS will benefit from using toxins that rapidly break up dead cells and thereby allow the weapon to keep injecting toxin into viable cells.

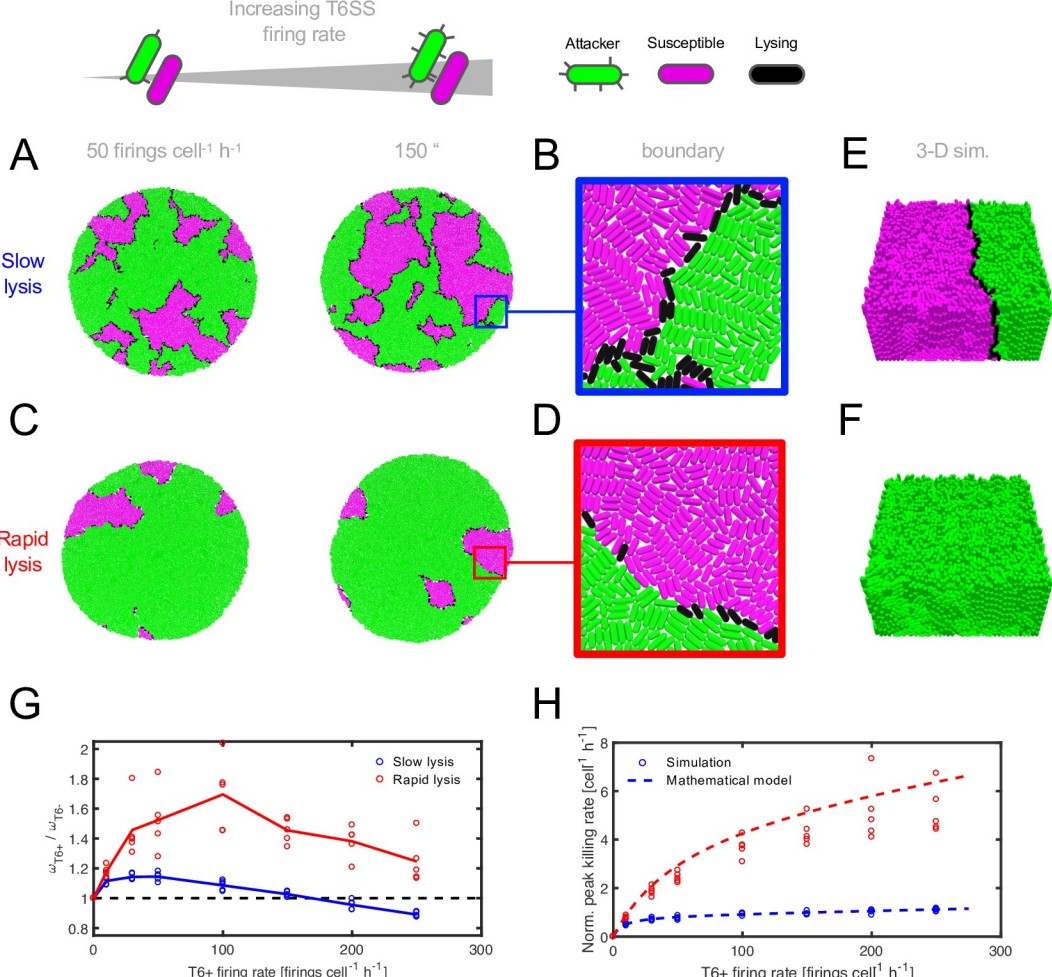

**Fig 1. Modeling predicts that victim lysis rate determines T6SS killing efficiency. (A, C)** Simulation snapshots compare attacker versus susceptible strain competition outcomes (see legend) for T6SS firing rates 50 and 150 firings cell$^{-1}$ h$^{-1}$ and for slow and rapid victim lysis. (**B, D**) Magnified sections of simulated communities showing occupation of interstrain boundary by lysing cells for the higher firing rate case shown in (A, C). (**E, F**) Snapshots of 3D competition simulations ("sim.") after 13 hours' growth, for slow (E) and rapid (F) victim lysis ($k_{fire}$ = 100.0 firings cell$^{-1}$ h$^{-1}$). (**G**) Parameter sweep measuring relative fitness of the T6+ attacker strain ($\omega_{T6+}$ / $\omega_{T6-}$, equivalent to ratio of strain division rates) for increasing attacker firing rate; solid lines denote means. (**H**) Comparison of normalized ("Norm.") peak killing rates from simulations (circles) with those predicted from the fraction of the interstrain boundary occupied by lysing cells (the "boundary saturation"; dashed lines, see Materials and methods). Parameter values: $N_{hits}$ = 1, $c$ = 0.001; "slow" and "rapid" victim lysis correspond respectively to $k_{lysis}$ = 0.8, 8.0 h$^{-1}$ throughout. Five simulation replicates are shown for each parameter combination; additional parameter values and quantification of boundary saturation are shown in S1 Fig and S2 Fig. Raw data are available at dx.doi.org/10.6084/m9. figshare.11980491. T6SS, type VI secretion system.

## Microfluidic experiments reveal a quantitative fit to modeling predictions

In order to test the model's predictions, we turned to microfluidic chambers, which enable single-cell imaging and real-time analysis of T6SS activity, interstrain boundaries, and victim lysis (Fig 2, S3 Fig, S4 Fig and S5 Fig). We competed a T6SS+ attacker strain *A. baylyi* against a T6SS− susceptible strain of *E. coli* [29]. Normally *A. baylyi* uses its T6SS to inject a cocktail of at least 5 different "effector" toxins whose molecular targets and activities differ substantially (see S5 Fig and S4 Movie for competition between parental *A. baylyi* strain with all 5 toxins and *E. coli*). Importantly, it is possible to engineer *A. baylyi* strains that carry only single-

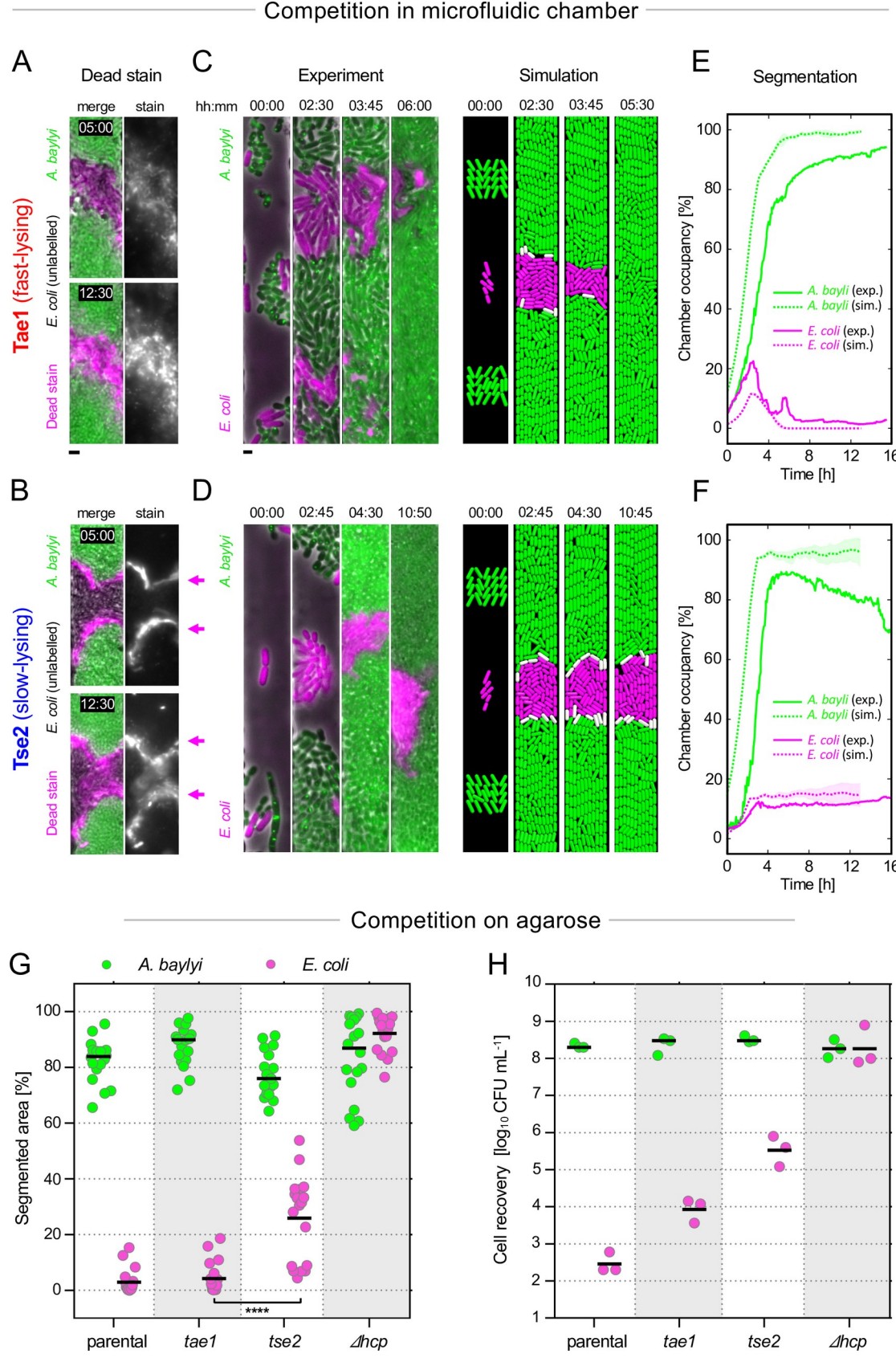

**Fig 2. Attackers with lytic T6SS effectors outperform nonlytic analogs in microfluidic and agarose competition assays.**
Comparison of T6SS competition dynamics in microfluidic chambers, varying T6SS effector type: T6SS+ attacker *A. baylyi* with lytic effector Tae1 (**A, C, E**) or nonlytic effector Tse2 (**B, D, F**). (**A, B**) Fluorescent microscopy images ("Dead stain") show unlabeled *E. coli* victim cells between 2 groups of T6SS+ *A. baylyi* (green) within observation channels. Propidium iodide dead stain (magenta) labels DNA released from lysed cells (Tae1, A), or DNA inside cells upon membrane permeabilization (Tse2, B). (**C, D**) Fluorescent microscopy time-lapse series ("Experiment," left column) show dynamics of competition (*A. baylyi*, green; *E. coli*, magenta); images are representative of 16 biological replicates. Simulations of chamber competitions using the agent-based model from Fig 1 are shown alongside ("Simulation," right column; attacker, susceptible, and lysing cells shown in green, magenta, and white, respectively). (**E, F**) Each strain's channel occupancy was measured in 5-minute intervals based on fluorescence signal and plotted (*E. coli* in magenta, *A. baylyi* in green) as percentage of the whole chamber (solid lines, "exp."). Data correspond to a single representative replicate; additional replicates are shown in Fig 3E and S3 Movie. These data are shown alongside analogous plots for chamber simulations (dashed lines, "sim"; lines and patches, respectively, denote means and standard deviations of 5 simulation replicates). Scale bars: 2 μm. (**G, H**) Competition assays carried out on agarose plates for same attacker and victim strains as in A–F, along with parental and T6SS-knockout ($\Delta hcp$) controls. Percentage area occupancies of each strain (**G**) were computed from fluorescence micrographs (S5 Fig). Two-way ANOVA ($\alpha = 0.01$) with Tukey post hoc test; **** $p \leq 0.0001$; $n = 18$ spot competitions analyzed per group. (**H**) Cell recovery data for these assays quantify *A. baylyi* and *E. coli* survival for the same 4 treatment groups, subsample of 3 replicates per case. Raw data are available at dx.doi.org/10.6084/m9.figshare.11980491. CFU, colony-forming unit; exp., experiment; sim., simulation; hcp, hemolysin-coregulated protein; T6SS, type VI secretion system; Tae1, type VI amidase effector 1; Tse2, type VI effector 2.

effector toxins while maintaining the same T6SS assembly rate [29]. This allows us to test our predictions by comparing the killing abilities of single-effector mutants bearing fast-lysing (Fig 2A, 2C and 2E) and slow-lysing (Fig 2B, 2D and 2F) toxins, during T6SS competition.

Fluorescently tagged strains of *A. baylyi* (green), expressing either toxin Tae1 (type VI amidase effector 1) or toxin Tse2 (type VI effector 2), were compared in competition with unlabeled T6SS-sensitive *E. coli* bacteria (Fig 2A). Tae1 is an amidase toxin that targets victims' cell walls and induces rapid (<1 minute) lysis following intoxication (Fig 2A), whereas toxin Tse2 (Fig 2B) lyses target cells slowly (>4-hour lysis delay)—yet still kills cells efficiently through a mechanism yet unknown [29]. *A. baylyi* cells carrying only Tae1 were able to largely clear the chamber of *E. coli* cells within 6 hours of inoculation (Fig 2C, see S3 Fig and Materials and methods for quantification procedure). In contrast, *A. baylyi* secreting only the slow-lysing toxin Tse2 (Fig 2D) were unable to clear *E. coli* from the chambers even after up to 18 hours' coincubation, despite efficiently killing cells on the boundary (S3 Movie shows 5 pairwise comparisons of the 2 toxins). We also examined the dynamics of *E. coli* killing in the presence of a propidium iodide fluorescent "dead stain": by counting the rates at which new propidium iodide foci appeared, we were able to estimate the *E. coli* death rate in the 2 treatments (S4 Fig and S5 Movie). These analyses confirmed that kill rates saturated earlier and were slower overall for the slow-lysing effector Tse2 compared with Tae1.

The microfluidics support the general prediction that causing rapid cell lysis greatly improves the functioning of the T6SS. To further test the fit between the model and the data, we ran a new version of our T6SS agent-based model using the specific geometry of the microfluidic system and compared the dynamics of killing between the model and the experiments. Importantly, this exercise was based on the same parameters as our earlier models and was not carried out by simply fitting the model to the data using free parameters in the model (see Materials and methods for details of parameter choice). Running the new model revealed a good fit between the predicted dynamics and those seen in the experiments, for both the low and the high rate of cell lysis (Fig 2E and 2F, S5 Fig and S2 Movie; see also Fig 3E for additional replicates and statistical analysis). This close quantitative fit with the data provides strong support for the model and the prediction that T6SS-dependent elimination of susceptible cells depends critically on the rate of target cell lysis.

Our microfluidic system enables observation of T6SS competition dynamics at single-cell resolution, allowing a direct comparison to our models. However, these experiments also confine bacteria in narrow chambers in a manner that may influence the outcome of experiments.

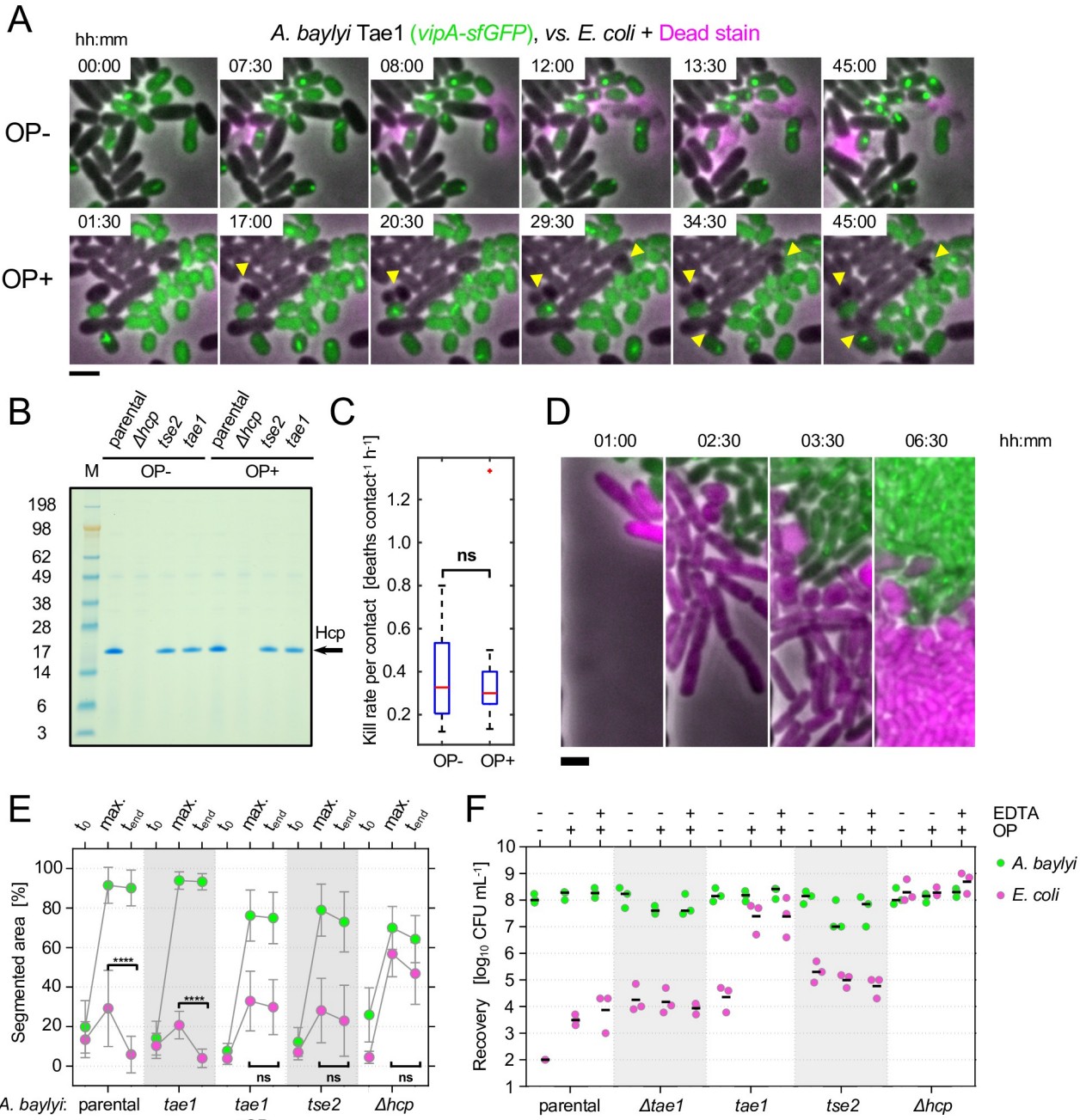

**Fig 3. Osmoprotective conditions prevent victim cell lysis and killing without affecting toxin activity. (A)** Time-lapse images show T6SS firing and killing dynamics between cells grown on agarose pads in presence and absence of osmoprotectant (OP+, OP−, respectively). Propidium iodide ("Dead stain" magenta) labels DNA released from susceptible *E. coli* (unlabeled) upon T6SS attack from *A. baylyi* cells (*vipA-sfGFP*, green) armed with toxin Tae1. Arrowheads mark formation of spheroplasts upon T6SS intoxication (PI does not label intoxicated cells under the OP+ condition). **(B)** Hcp was precipitated from culture supernatants of *A. baylyi* with and without OP, with proteins separated by SDS-PAGE and stained using Coomassie Blue. Column M contains protein reference ladder; axis values show approximate protein molecular weight in kDa. **(C)** Image analysis of time lapses comparing *E. coli* death rates, normalized by *E. coli*–*A. baylyi* initial contact counts, in the presence and absence of OP (2-group *t* test; *p*-value = 0.9027; confidence interval for difference in means $-0.288 < \mu_{OP+} - \mu_{OP-} < 0.256$). Data originate from ten 45-minute time lapses as in (A), also shown in S6 Movie. Boxplot shows data ranges (dashed lines), interquartile ranges (blue boxes), and means (red lines). **(D)** Fluorescence time-lapse series of microfluidic competition assay between *A. baylyi* armed with Tae1 and *E. coli* in the presence of OP at indicated time points (hh:min). Images are representative for 16 biological replicates. **(E)** Quantification of chamber dynamics plot initial ($t_0$), maximum ($t_{max}$), and final ($t_{end}$) area occupancies of *E. coli* (magenta) and *A. baylyi* (green) from microfluidic competition assays. Data for indicated *A. baylyi* strains and treatments are displayed. Two-way ANOVA ($\alpha = 0.01$) with Tukey post hoc test; ****$p \leq 0.0001$; $n = 16$ channel competitions were analyzed per group. **(F)** *A. baylyi* and *E. coli* recovery data for agarose competition experiments are shown in presence (+) or absence (−) of OP. For each OP+ condition, we

show an additional treatment with 20 mM EDTA added to ensure lysis of spheroplasts pre-recovery. Scale bar: 2 μm. Raw data are available at dx.doi.org/10.6084/m9.figshare.11980491. CFU, colony-forming unit; exp., experiment; Hcp, hemolysin-coregulated protein; ns, not significant; OP, osmoprotectant; sfGFP, super-folding green fluorescent protein; T6SS, type VI secretion system; Tae1, type VI amidase effector 1; Tse2, type VI effector 2; VipA, ClpV-interacting protein A.

We therefore sought also to test our predictions by coculturing cells using the agarose pad assay widely used in T6SS studies (Fig 2G and 2H; Materials and methods). Micrographs and cell recovery assays revealed large numbers of surviving *E. coli* microcolonies when competed with *A. baylyi* secreting only slow-lysing Tse2 but little *E. coli* survival when incubated with *A. baylyi* secreting Tae1 (Fig 2G and 2H, S6 Fig). Specifically, approximately 30 times more *E. coli* cells survive in the presence of effector Tse2 than with effector Tae1 (Fig 2H).

## Slowing cell lysis with osmoprotectant suppresses T6SS killing

Both in microfluidic chambers and on agarose surfaces, the fast-lysing T6SS toxin Tae1 outperforms its slow-lysing counterpart Tse2. Moreover, we observe the buildup of large clumps of victim cells in the absence of cell lysis, as predicted by the corpse barrier effect. These findings, which compare 2 naturally occurring toxins, support our model's predictions. However, if the toxins kill at different rates, this may also influence the competition outcomes in addition to the effects of lysis. We therefore devised a second experimental strategy to modulate victim lysis without changing the toxin secreted by the attacker. We reasoned that, because Tae1 causes cell lysis by degrading peptidoglycan (Fig 2), it ought to be possible to prevent this by the addition of osmoprotectant (sucrose and $MgSO_4$), which would stabilize the intoxicated cells as spheroplasts [38,39].

As anticipated, in the presence of osmoprotectant, *E. coli* cells incubated with Tae1-armed *A. baylyi* blebbed and formed spheroplasts with intact membranes, contrasting the rapid lysis without osmoprotectant (Fig 3A and S6 Movie). Importantly, the presence of the osmoprotectant had no detectable effect on the T6SS activity of *A. baylyi*. With and without osmoprotectant, protein precipitation and SDS-PAGE suggested equal amounts of secreted hemolysin-coregulated protein (Hcp, a structural component of T6SS needles) in the culture supernatant (Fig 3B). Furthermore, image analysis did not identify any significant difference in the rates of *E. coli* cell killing, based on cell lysis or blebbing events per number of contacts between cells of the 2 species (Fig 3C). This shows that osmoprotective conditions specifically block target cell lysis without altering T6SS activity and potency or delivery of toxin Tae1. Strikingly, the addition of osmoprotectant resulted in outcomes that were very similar to competitions with the slow-lysing effector Tse2 (Fig 2B), with a visible corpse barrier of spheroplasts forming at the *A. baylyi*–*E. coli* interface (Fig 3D and S7 Movie) and the inability to clear prey cells in the microfluidic device (Fig 3E).

The same outcome was also seen on agarose, with osmoprotection greatly increasing *E. coli* survival when incubated with *A. baylyi* secreting fast-lysing toxin Tae1. However, osmoprotection had no effect on *E. coli* survival in mixtures with *A. baylyi* secreting the slow-lysing toxin Tse2. Here we added a treatment in which the membrane-destabilizing ion chelator EDTA (see Materials and methods) was added during cell recovery, to confirm that osmoprotectant did not enable cell survival on the recovery plates and so alter the killing effectiveness of the toxin (S7 Fig), which it did not (Fig 3F). Finally, we showed that osmoprotectant has no effect on killing by an *A. baylyi* strain engineered to lack only Tae1 (Fig 3F), which confirms that there are no unintended effects of osmoprotectant and that it only impacts the cell wall targeting effector Tae1.

For 2 independent methods of controlling target cell lysis rate—and for both microchamber and agarose assays—we see that T6SS weaponry performs much better when it is able to rapidly lyse target cells. However, these experimental results are still limited to 2 T6SS effectors from a single bacterial species. To perform a broader assessment of the value of lytic toxins across many species, therefore, we turned to genomic data on the T6SS effectors found across Proteobacteria.

## Lytic toxins are extremely common across T6SS-wielding bacteria

Our model suggests that natural selection should drive a strong association between the use of the T6SS and delivery of effectors that disintegrate victims and clear the way for new targets. The model does not predict that the T6SS will always deliver lytic toxins; lysis is less important when cells of different genotypes are typically well-mixed or if the function of the T6SS is to kill the occasional cell that lands upon an existing community. Nevertheless, we see a clear benefit to lysis under a wide range of conditions. In order to look for an association between the T6SS and the use of lytic toxins, we reanalyzed genomic data [40] across 466 bacterial species (arranged in a cladogram in Fig 4), dividing effectors into those more likely to promote rapid cell lysis (peptidoglycan- and membrane-targeting) and those more likely to cause little or delayed lysis (pore formers, nucleases, and glycohydrolases; see Materials and methods for further discussion).

Of the 1,125 effectors identified in this dataset, 83.2% are predicted to cause rapid lysis (936/1,125), with 85.0% of strains analyzed carrying at least 1 fast-lysing effector (396/466). The structure of the taxonomy tree suggests that this abundance cannot be attributed to phylogeny alone, because species within the same clade are not especially similar compared with random species pairings (see Materials and methods and S8 Fig). For some effectors, there is appreciable phylogenetic signal at the genus level, but this rapidly falls away for broader taxa, showing that the data are not explained by a few major clades with particular effectors dominating the tree. This pattern is also not explained by bacteria showing a general preference for fast lysing toxins to kill competitors: the best-studied set of bacterial toxins, the colicins, are released to diffuse in the environment, and there, DNase and pore-forming mechanisms of action predominate [41,42]. Moreover, a second, rarer, contact-dependent mechanism, contact-dependent growth inhibition (CDI), appears to often deliver nonlytic toxins [43,44]. Although less studied, CDI is thought to function in different ways to the T6SS [45], with a major putative function being signaling within a genotype in addition to the elimination of other genotypes [46]. In sum, insofar as one can predict the speed of target lysis from an effector's mode of action, the prevalence of fast-lysing T6SS effectors across Proteobacteria is consistent with our prediction that causing cell lysis is often important for T6SS function.

Although we focused on strains with single effectors here, it is common for the T6SS to simultaneously carry multiple different effectors. If one lytic effector can be so potent, why carry so many other toxins? One possibility is that using a combination of effectors leads to more rapid intoxication and/or lysis [40,47]. Consistent with this, although the lytic effector Tae1 is highly effective when used in isolation, wild-type *A. baylyi* cells that carry all 5 naturally occurring effectors on their T6SS are even more effective killers (Fig 3F). Another likely source of toxin diversity is negative frequency–dependent natural selection, which arises because bacteria are more likely to carry resistance to the most common toxins. This can favor the use of rare toxins or combinations of toxins to make full resistance in target cells unlikely [2,48].

## Conclusions

The prevalence of the T6SS is testament to its importance in microbial ecosystems, in which it is a mediator of interbacterial competition across a broad range of contexts, including plant-

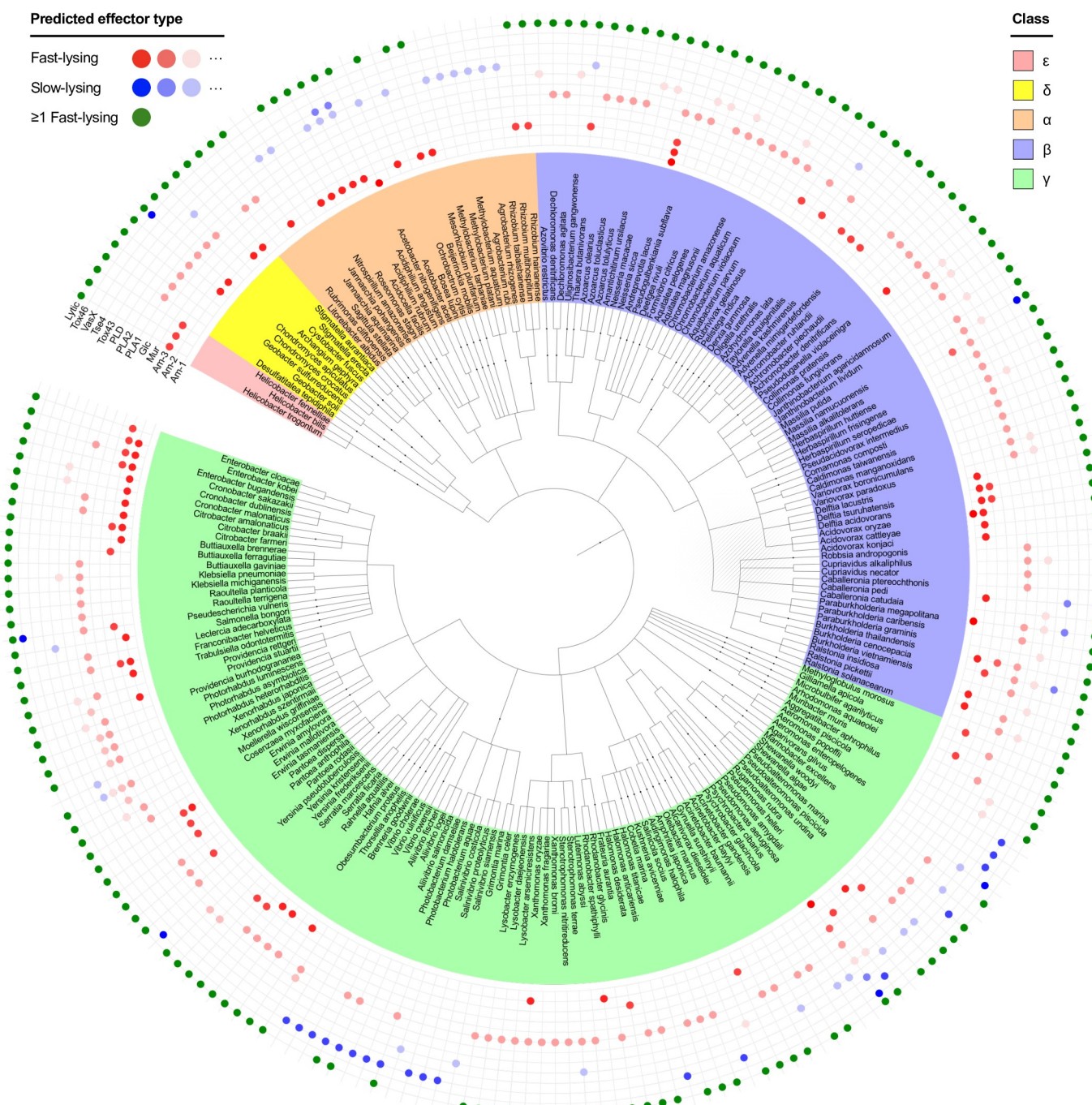

**Fig 4. Taxonomy showing distribution of lytic and nonlytic T6SS toxins across the Proteobacteria.** Each species is positioned in the tree according to its taxonomic classification in the NCBI taxonomy database and colored according to Proteobacterial Class (α, β, δ, ε, γ). Colored circles indicate the T6SS effectors associated with a single, randomly chosen strain from each species. Effectors are labeled according to their molecular target (see below); effectors expected to induce rapid cell lysis (Am-1-3, Mur, Glc, PLA1-2, PLD; "fast-lysing") are shaded in red; those expected to lyse cells more slowly (Tox43, Tse4, VasX, Tox46; "slow-lysing") are shaded blue. A green marker signifies that a given strain has at least 1 fast-lysing effector in its repertoire. Each species in this dataset is represented by 1 strain only, and so these data are not representative of the full toxin repertoire of any given species. However, by comparing example strains across many species, one can robustly assess preference for known lytic toxins. Plotting data from all 474 species was not feasible in 1 figure, so we plot approximately half (222) of the strains here, retaining at least 1 representative from each genus. The full tree is available to download as a separate file (S1 Data). Original data are from LaCourse and colleagues [40]; reanalyzed data are available from dx.doi.org/10.6084/m9.figshare.11980491. Am-1-3, peptidoglycan amidases; Mur/Glc, peptidoglycan glucosidases; NCBI, National Center for Biotechnology Information; PLA1-2 and PLD, phospholipases; Tox43, DNases; Tox46; NAD(P)+ glycohydrolases; Tse4/VasX, pore-forming.

and human-associated communities [11–16]. A much-discussed strength of the system is that it can directly deliver a wide range of toxins that would otherwise be unable to cross the membranes of target cells [2,20,49,50]. A corollary is that T6SS-wielding bacteria have the potential to serve as powerful probiotics, which deliver a chosen antimicrobial to harmful bacteria [51–53]. However, here we have shown that the T6SS is not simply a mechanism for delivering generic toxins to target cells; we find it can be ≥30 times less effective at clearing bacteria if unable to function as a lytic weapon (Fig 2H). This suggests that biotherapeutic strategies should consider the fit between each antimicrobial's mode of action with its mode of delivery, in a manner not seen with the use of conventional antibiotics. More generally, our work highlights the need to understand how bacterial weapons evolve if we are to harness them for our own ends.

## Materials and methods

### Agent-based modeling

S1 Fig provides an overview of our agent-based model, whose parameters and variables are summarized in S1 Table and S2 Table, respectively. The model is based on CellModeller (https://haselofflab.github.io/CellModeller/), a modeling framework designed by Tim Rudge and colleagues [25].

**Cell growth and division.** As in previous studies, our model represents bacterial populations as collections of rod-shaped cells, which divide via binary fission. We assume throughout that nutrients are always available in excess, so that all cells have access to the same nutrient concentrations. In this regime, each cell's volume $V_i$ increases exponentially through elongation, from birth volume $V_0$, according to the equation $dV_i/dt = k_{grow,i}V_i$; cell volumes are updated iteratively using the discretized form $\Delta V_i = k_{grow,i}V_i dt$, where $dt$ represents the simulation timestep and $k_{grow,i}$, the cell's growth rate. Cells divide lengthwise into 2 identical daughter cells once they reach volume $2V_0+\eta_{division}$, where $\eta_{division}$ represents uniform random noise in the cell cycle. Each daughter's axis vector $\hat{\boldsymbol{a}}_i$ is perturbed slightly by a noise term with weight $\eta_{orientations}$, to represent spatial imperfections in the division process.

**Cell movement.** Following the cell growth phase, the cell configuration is returned to a quasistationary mechanical equilibrium using an energy minimization algorithm, described in previous publications [25,26]. Briefly, any pair of cells whose surfaces are within 0.01 μm of each other are deemed to be overlapping and subject to mutual repulsion. Overlaps between neighboring cells are identified and summarized in a contact matrix $A$ and a distance vector $\boldsymbol{d}$, along with a regularizing matrix $M$ representing the energetic cost of cell movement, weighted by a scalar factor $\alpha$. Cell impulses $\boldsymbol{p}$ satisfying the equation $(A^T A+\alpha M)\boldsymbol{p} = -A\boldsymbol{d}$ are calculated using an iterated conjugate gradients method [54], such that the application of the impulses moves the cells back to an equilibrium configuration while minimizing cell displacement, to within an absolute tolerance $\epsilon_{CG}$. New overlaps created by movement are resolved sequentially, adding sets of impulses together until either their application produces no additional overlaps or until the iteration count exceeds the maximum iteration number $M_{iter,max}$.

**T6SS firing and costs.** We model T6SS activity in terms of discrete, spatially explicit firing events (See S1A Fig, "cells"). Every simulation timestep $dt$, each T6SS+ cell $i$ fires $N_{firings,i}$ times, with $N_{firings,i}$ being drawn at random from a Poisson distribution with mean $k_{fire}$ (S1B Fig). For each of these firings, a point $\boldsymbol{x}$ on the surface of cell $i$ is chosen at random. A vector with origin $\boldsymbol{x}$, orientation $\hat{\boldsymbol{u}}(\boldsymbol{x})$, and length $L_{needle}$ is constructed, where $\hat{\boldsymbol{u}}(\boldsymbol{x})$ corresponds to the unit outward normal vector at point p on the cell. We assume that $L_{needle} = R$ (the cell radius), based on the observations that (1) extended T6SS sheaths often span the diameter of producing cells and that (2) contraction roughly halves the sheaths' length [55–57]. We assume

firing to be constitutive (fixed $k_{fire}$) and rapid compared with cell movement timescales, such that firing occurs in essentially static cell configurations. We assume a linear relationship between T6SS firing rate and growth costs (S1C Fig): firing $N_{firings,i}$ times on a given timestep $dt$ reduces the growth rate of cell $i$ by factor $1-c_{Total,i}$, where cost $c_{Total,i} = c(N_{firings,i}/dt)$.

**T6SS hit detection and response.**   Following firing, each T6SS needle vector is checked to see if it passes through any other cell in the current configuration (a needle never strikes the cell that produced it). Geometrically, this is equivalent to checking whether 2 line segments come within distance $R$ of one another, where $R$ is the cell radius. This computationally intensive process is accelerated using spatial sorting algorithms and parallel implementation, described previously for computing cell–cell mechanical interactions [25]. Tallies of successful needle hits are kept to track each cell's intoxication. We assume a stepwise toxin response (S1D Fig) with cells dying after being struck by $N_{hits}$ needles. To model immunity, a separate tally is kept for each type of cell in the simulation. For example, given 2 mutually susceptible T6SS+ strains X and Y, X would ignore its tally of hits from other X cells (self-immunity) but respond to those from Y cells.

**Model parameterization.**   S1E Fig provides an overview of the parameters governing T6SS behavior in our model. There are 17 parameters altogether, whose names and values are summarized in S1 Table. Mechanical and numerical parameters, governing cell movement during growth, were taken from previous publications. Whenever possible, parameters governing T6SS firing and response were estimated directly from experimental observations of *A. baylyi*/*E. coli* competition. This was not possible in the case of the cost parameter $c$, and so we performed a broad parameter sweep to test its influence in our competition simulations from Figs 1 and 2. S1F Fig summarizes this parameter sweep, plotting final attacker frequency, (*Total attacker cell volume* / *Total cell volume*), as a metric of competition outcome. Following this sweep, we set the cost factor to the intermediate value of 0.001, for which the optimum firing rate (i.e., the $k_{fire}$ value maximizing final attacker frequency) approximately coincides with the observed *A. baylyi* firing rate (50 firings cell⁻¹ h⁻¹; details next). For simulations of the microfluidic device, the majority of parameters remained unchanged from our earlier models ($N_{hits} = 1, c = 0.001, k_{lysis} = 0.8, 8.0$ h⁻¹). Firing rate $k_{fire}$ is an independent variable in our models, and we set this to 50.0 firings cell⁻¹ h⁻¹, based on *A. baylyi's* firing rate under these conditions. The 1 model parameter that, in a sense, was fitted was the degree of mechanical growth restriction (the degree to which dense cell packing causes a slowing in growth), which was increased moderately ($1/\gamma = 0.1$). We found this was necessary to stop susceptible cells from rapidly pushing attackers out of the openings of the chamber in the new geometry.

### Simulation protocols

**Computation and postprocessing.**   Agent-based model simulations were run on a 2017 Apple MacBook Pro laptop computer, with simulations distributed between Intel 3.1 GHz quadcore i7-7920HQ CPU, Intel HD 630 Graphics card, and AMD Radeon Pro 560 Compute Engine. Simulation data were analyzed using custom Matlab scripts and visualized using Paraview [58].

**2D disc simulations (Fig 1).**   Here we used 100-cell inocula consisting of a 1:1 mix of T6SS+ "attacker" and T6SS− "susceptible" cells, randomly scattered and oriented within a 100-μm circle. Cell coordinates were restricted to a 2D plane, producing a confluent monolayer of cells approximately 180 μm in diameter. Simulations were set to terminate once the cell population exceeded 10,000 individuals (living or dead), representing an ecological niche with limited space.

**3D "biofilm" simulations (Fig 1).**   Here we instead began with 1 cell of each type but allowed cells to move and rotate freely within a walled box with base dimensions 40 by 40 μm. To model cell detachment from the mature biofilm, a cell "slougher" was added to remove cells positioned >20 μm from the biofilm's base. Simulations were set to terminate after 13 hours of growth.

**Simulations of microfluidic chambers (Fig 2).**   In this case, we grow cells from the initial cell arrangement shown in Fig 2C ("Simulation," t = 00:00), in walled, open-ended chambers measuring 10 by 100 μm, with cells being removed from the simulation after being forced out of the chamber's ends. Chamber simulations began with a fixed arrangement of cells, created by placing a susceptible cell in the center of the chamber, flanking it with 2 rows of 5 attacker cells, and then allowing 3 hours of growth. Simulations terminated after 13 hours of growth following this starting point, shown in S2 Movie as t = 0.0 hours.

## Simulation metrics

**Relative fitness.**   We used the relative cell division rate of the T6SS+ strain, $\omega_{T6+}/\omega_{T6-}$, to quantify simulation outcome. This quantity is defined as

$$\frac{\omega_{T6+}}{\omega_{T6-}} = \frac{log_2\left(V_{total,T6+}(t_{end})/V_{total,T6+}(t_{start})\right)}{log_2\left(V_{total,T6-}(t_{end})/V_{total,T6-}(t_{start})\right)},$$

where $V_{total,T6+}$ and $V_{total,T6-}$ correspond to the total volumes of T6+ (attacker) and T6− (susceptible) cells, respectively, and $t_{start}$, and $t_{end}$ are the simulation start and end times, respectively.

**Interstrain boundaries.**   We define T6SS+ cells as lying on the interstrain boundary if they are touching and therefore within T6SS firing range of any T6SS− susceptible cell, whether living or dead (see S2B Fig). As shown in S2A Fig, the number of boundary cells, $N_{boundary}$, varies both within and between simulations, and so it is important to normalize out this variation when comparing T6SS kill rate values, as discussed next.

**Boundary saturation.**   To quantify the proportion of interstrain boundaries occupied by lysing cells ("boundary saturation," see Fig 1), we identified attacker cells as lying on the interstrain boundary (bT6SS+) if they touched at least 1 T6SS− cell, either living or lysing, and then calculated the fraction of all bT6SS+ | T6SS− contacts that involved dead cells. Plotting these values as a function of simulation time (S2I Fig, left column, blue circles) shows that, as the competition progresses, interstrain boundaries become increasingly populated with lysing cells—i.e., boundary saturation increases. As in Fig 1, raising the firing rate (S2I Fig, middle and right columns) substantially increased the final saturation, to the point that >95% of all susceptible cells within range of T6SS attacks were already dead. Conversely, increasing the victim lysis rate reduced the average saturation at all time points and for each firing rate (S2I Fig, red circles). For each of these cases, the corresponding simulation snapshots are shown in S2G Fig and S2H Fig.

**Confluency.**   Analysis of cell–cell contacts can also be used to define the point at which disc colonies become confluent. At this time, most cells become surrounded on all sides, and so the median cell coordination number (i.e., the median number of neighbor cells in physical contact with a focal cell; S2C Fig) plateaus.

**Normalized peak kill rate.**   At a given simulation time point $t$, we compute the net victim death rate $k_{death}(t)$ as the gradient of the cumulative susceptible cell death count $N_{death}(t)$: $k_{death}(t) = (dN_{death}(t)/dt)_t$. S2D Fig shows plots of both $k_{death}(t)$ (red) and $k_{death}(t)$ (blue) for reference. In Fig 1, we use $max(k_{death}(t_{max}))$, normalized by the number of boundary cells at the corresponding time point, $N_{boundary}(t_{max})$, as a measure of the postconfluency kill rate per unit

interstrain boundary,

$$Normalized\ peak\ killing\ rate = max(k_{death}(t_{max}))/N_{boundary}(t_{max}).$$

Computed in this way, normalized peak T6− kill rates function as an intrinsic metric of T6SS effectiveness: plotting these rates against T6SS firing rate (S2E Fig) produces the same saturating curve for different weapon costs. Note that this is not equivalent to taking $max(NKR) = max(k_{death}(t)/N_{boundary}.)$: as shown in S2D Fig (inset), this would give noisy kill rate values representative of preconfluent killing instead of a converged kill rate that incorporates barrier effects.

**Predicting kill rate from boundary saturation.** To test whether boundary saturation could be used to predict T6SS killing rate, we compared peak-time saturation values (see S2I Fig) with peak-time killing rates, as plotted in Fig 1H. We found that boundary saturation could be used to predict T6SS killing using an intuitive formula, $k_{kill} = (1-f_{boundary})k_{fire}\,p_{hit}$, where $k_{kill}$ is the rate of victim cell death per unit interface, $f_{boundary}$ is the boundary saturation, $k_{fire}$ is the attacker firing rate, and $p_{hit}$ is the probability of a successful T6SS attack on the boundary (see next).

**T6SS hit probability.** To compute the probability that an attacker cell (situated on an interstrain boundary) hits a nonkin cell with a T6SS needle, we used our model to simulate rounds of firing in fixed cell configurations. For each of the 2D simulations shown in Fig 1, the final cell configuration was extracted. Boundary attacker cells were identified (see "Interstrain boundaries" previously; S2B Fig) and allowed to fire $N$ needles at random. Needle hits were computed and the proportion of needles striking nonkin (susceptible or victim) cells calculated and averaged over the set of configurations. These mean values were found to be invariant with respect to the firing rate used to generate the input cell configurations. The data reported in the main text were measured for simulations with $k_{fire} = 50.0$ firings cell$^{-1}$ h$^{-1}$, $N = 100$.

## Bacterial strains and cultivation

All strains were grown at 37˚C on LB (lysogeny broth) agar plates or by shaking at 200 rpm in LB broth. Growth media were supplemented with appropriate antibiotics where indicated. To gentamycin-resistant *E. coli* MG1655 and isogenic mutant constitutively expressing mRuby3 derivate, 15 μg/mL gentamicin was added; for *A. baylyi* ADP1 rpsL-K88R derivatives, 50 μg/mL streptomycin was added. Mutagenesis in *A. baylyi* was carried out as described previously [29]. The strains used in this study are listed in S3 Table.

## Microfluidics

A custom-built polydimethylsiloxane-based microfluidic device, similar to that used in a previous study [59], was used for all microfluidic experiments. As shown in S3A Fig, 2 individual flow channels are linked by narrow observation channels 10 × 30 to 150 × 1 μm in dimension (width × length × height). This design allows observation of competition dynamics in cell monolayer, as clonal groups of bacteria grow and come into contact. During design iterations, we realized that *A. baylyi* is slightly thicker than *E. coli* and so was prone to be trapped at observation channel entrances, whereas *E. coli* cells could readily enter the channels. On the other hand, *E. coli* grows faster than *A. baylyi* and so could potentially expel the latter from chambers, interfering with the outcome of the contact-dependent competitions. Adjustments to the loading procedure helped to overcome this problem: First, we loaded *A. baylyi* by applying a higher flow rate to the top flow channel (top channel 0.001 μL/second, bottom 0.003 μL/s for 30 seconds), thereby forcing *A. baylyi* cells into the tops of the observation channels. Next, *E.*

*coli* cells were loaded from the opposite side using the same principle but with inverted flow pressures. Most *E. coli* cells therefore became trapped inside the central regions of the observation channels. Last, the bottom cell inlet was exchanged from *E. coli* to *A. baylyi* cells, allowing us to load a second layer of *A. baylyi* to block the exit of the observation channel (S3A Fig). Using this procedure, a prey cell population was trapped in between 2 layers of *A. baylyi* cells. The chip was perfused with LB medium at 0.005 μL/second using a Nemesys syringe pump and incubated at 30°C inside an Oko-lab incubation chamber. Competition experiments were run for up to 18 hours.

Cells were imaged on an inverted Nikon Ti Eclipse epifluorescence microscope equipped with a fully motorized stage and perfect focus system for multiposition time-lapse imaging. Images were acquired at a 5-minute acquisition frame rate using a 1.42 numerical aperture Plan Apo 100× oil immersion objective. Fluorescence was excited using a SPECTRA X light engine and filtered using ET-GFP (Chroma #49002) and ET-mCherry (Chroma #49008) filter set and recorded on a pco.edge 4.2 (PCO, Kelheim, Germany) scientific complementary metal-oxide-semiconductor (sCMOS) camera (pixel size 65 nm) using VisiView software (Visitron Systems, Puchheim, Germany). For all excitations, output power of the SPECTRA X light engine was set to 20%. Phase contrast, GFP, and propidium iodide were recorded at 100-millisecond exposure, whereas mRuby3 was recorded at 250-millisecond exposure.

Recorded time-lapse series were postprocessed with Fiji [60]. Stage drift was corrected using a customized StackReg plugin [29]. Observation channels were selected manually based on initial cell density (<40% occupancy) and a 2:1 (*A. baylyi*:*E. coli*) cell ratio. Next, an automated macro was generated to split fluorescence channels, normalize single intensity to 0.1% saturated pixels per frame, and reduce noise by blurring images using a Gaussian blur filter (sigma: 2 pixels). If necessary, mRuby3 signal background fluorescence was subtracted (50-pixel rolling ball radius, sliding paraboloid, disabled smoothing). For signal segmentation, a global Otsu thresholding algorithm was applied to plot growth trajectories for each strain. The total occupancy of *A. baylyi* + *E. coli* at confluency (assessed from phase-contrast image) served as an internal quality control for segmentation (S3B Fig), which was not allowed to be higher than 110% (10% error above expected 100% *A. baylyi* + *E. coli* channel occupancy).

## Microbial competition assays

Overnight cultures of *A. baylyi* and *E. coli* were diluted 1 to 20 and 1 to 100, respectively, into fresh LB and grown to an optical density (OD, measured at 600 nm) of 1. *vipA-sfGFP*-labeled *A. baylyi* strains served as a T6SS-positive parental strain, whereas an isogenic *hcp* deletion mutant was used as T6SS-negative control. *E. coli* served as the T6SS-susceptible strain; attacker and susceptible cells were concentrated to indicated ODs and mixed at a 1:1 ratio. Five microliters of cell mixtures was spotted on predried LB plates. After spots were completely absorbed, the competition was carried out for 3 hours at 37°C. Subsequently, spots were excised from LB plates and bacterial cells were resuspended in 500 μL LB and subjected to 7 rounds of 10-fold serial dilution. Colony-forming units (CFUs) of *A. baylyi* and *E. coli* cells were enumerated respectively by plating on streptomycin (100 μg/mL) or gentamycin (30 μg/mL) media, incubated at 37°C.

For low-magnification microscopy of spot competition assays, cells were grown and prepared as indicated previously. A total of 0.5 μL of cell mixture was spotted on a 2.5-mm-thick 1%-agarose (w/v) and LB pad and allowed to dry. The spots were imaged with a 10× (numerical aperture 0.25) air objective lens, using the same microscope setup as before. At this point, fluorescent signal from bacterial cells was too low to be reliably distinguished from background noise. The position was marked, and agar pads were incubated in a humidified

chamber (100% relative humidity) at 37°C for 3 hours. Subsequently, phase-contrast images, GFP (1-second exposure), and mRuby3 (2-second exposure) fluorescence images were acquired for the same positions. Images were background-corrected (50-pixel rolling ball radius, sliding paraboloid, enabled smoothing) and blurred (sigma radius 2 pixels). Fluorescent signal of both channels was segmented using a global Otsu thresholding algorithm in Fiji [60]. Percentage occupancy was reported for each strain; overlay masks of the segmented signal were generated and compared with the original image as a quality control.

## Hcp secretion assay

For detecting Hcp secretion into culture supernatant, indicated *A. baylyi* strains were regrown as described for the quantitative competition assay. Thereafter, 2 mL of fresh LB, or LB + osmoprotectant (see next), were inoculated at $OD_{600}$ 0.5 and incubated shaking at 37°C, 200 rpm for 2 hours. Subsequently, 1 mL of culture was centrifuged for 1 minute at 10,000$g$ and 4°C. After centrifugation, 900 μL of culture supernatant were transferred into a fresh tube and proteins were precipitated by adding 100 μL ice-cold 100% TCA (w/v; Sigma-Aldrich). The samples were incubated on ice for 10 minutes with regular mixing and then centrifuged for 5 minutes at 14,000$g$ and 4°C. The pellets were washed with ice-cold acetone, dried at room temperature, and then resuspended in 20 μL 1× NuPAGE LDS sample buffer (Thermo Fisher Scientific); 2 μL 1 M dithiothreitol was added and then incubated at 70°C for 10 minutes. Samples were loaded onto NuPAGE 4%–12% Bis-Tris 1.0-mm, 12-well protein gels (Thermo Fisher Scientific), which were run in MES buffer (Thermo Fisher Scientific) for 50 minutes at 150 V. The gels were stained with InstantBlue Coomassie protein stain (Expedeon) for 1 hour at room temperature and subsequently destained with distilled water. The results are representative for biological duplicates.

## Osmoprotection assay

Osmoprotectant (OP) was used to stabilize cells against lysis by the effector Tae1. *A. baylyi* does have other lytic effectors (e.g., the phospholipase Tle1), but these target cell membranes rather than the cell wall and were, as expected, unaffected by osmoprotective conditions (Fig 3F). Cell wall–damaged *E. coli* cells were stabilized as spheroplasts [38,61] by the addition of 0.4 M sucrose and 8 mM $MgSO_4$ (Sigma-Aldrich) to the growth medium. For Hcp secretion assays (Fig 3B), cells were pregrown in absence of osmoprotective medium. For microbial competition experiments, LB plates as well as recovery media were supplemented with OP. To prevent spheroplast regrowth after competition experiments, cells were recovered in LB-OP supplemented with 20 mM EDTA and incubated for 45 minutes at 37°C, which causes spheroplast lysis. This approach was based on experiments showing that EDTA overrides osmoprotective stabilization of spheroplasts formed using the antibiotic ampicillin (S7 Fig).

For comparison of T6SS intoxication rates in the presence or absence of OP, we competed unlabeled *E. coli* with *vipA-sfGFP*-labeled *A. baylyi tae1* single-effector mutant. Cells were grown and prepared as indicated for microbial competition assays. A total of 2.5 μL of cell mixture was spotted onto fresh 1%-agarose (w/v) in LB pads containing 2 μg/mL propidium iodide and covered with a glass coverslip and subsequently imaged for 45 minutes at 30°C at a 30-second acquisition frame rate using the same microscope setup as for microfluidic competition assay. From recorded time-lapse series, we manually counted T6SS intoxication events, logging any instance of cell lysis, blebbing, or other sudden morphological change as indicating successful intoxication by Tae1. To account for variation in *A. baylyi*/*E. coli* cell contact between treatments, we then normalized intoxication counts by the number of unique *A. baylyi*/*E. coli* cell contacts visible at the start of each time lapse. These measurements were repeated for all 10

time lapses; measurements taken in the presence (OP+) and absence (OP−) of OP were compared with a 2-sample $t$ test performed in Matlab.

## Determination of T6SS firing rate in *A. baylyi*

The rate of T6SS firing in *A. baylyi* was measured from 1,000 *vipA-sfGFP*, *clpV-mCherry2*-labeled cells via automated detection of ClpV foci, using Trackmate software [62]. Each new ClpV focus was assumed to mark 1 T6SS contraction event in a focal cell [29]. Time-lapse series were recorded for 7 minutes at a 10-second acquisition frame rate, at 30˚C, giving an average measurement of 0.8 firings cell$^{-1}$ minute$^{-1}$ (48 firings cell$^{-1}$ hour$^{-1}$).

## Survey of T6SS effector types across Proteobacteria

The dataset of LaCourse and colleagues [40] lists the distribution of 12 known T6SS toxin domains (Tae1-4, Tge1-3, Tle1-5, Tse4, and VasX effectors; Tox46 and Tox34 motifs) across the genomes of 466 bacterial species, spanning the phylum Proteobacteria. Each species is represented by a single randomly chosen strain, and so the effectors listed for any 1 strain are not necessarily representative of the effector variety seen within that species. We used the NCBI common tree tool to arrange these strains into a cladogram, according to their taxonomic classification in the NCBI taxonomy database [63]. We then annotated each strain according to the effectors found in its genome, categorizing each effector as "fast-lysing" or "slow-lysing" based on its molecular target: peptidoglycan, phospholipids, DNA, membrane pore-forming, and NAD(P+).

For our purposes, "lysis" denotes a loss of target cell integrity, such that it ceases to be a physical barrier to the movement of encroaching T6SS+ attackers. Given this definition, we considered how effectors' modes of action might affect the timescale of lysis. The peptidoglycan cell wall is the key structure in the bacterial cell for resisting turgor pressure, and its disruption is known to cause rapid lysis [29,64–67]. Consistent with this, we found that delivery of the amidase toxin Tae1 rapidly induces cell lysis (Fig 2 and S6 Movie), <1 minute after intoxication. Disruption to cell membranes can also lead to lysis, particularly for phospholipid-targeting toxins that break down the membrane itself [29,68,69]. This contrasts with pore-forming toxins, which function by inserting themselves into intact membranes. This allows solutes to diffuse across the membranes and disrupt the proton-motive force but without rapidly reducing the structural integrity of the cell [47,70]. Similarly, nuclease toxins, which degrade genetic material in the cytosol, are expected to disable bacterial targets without immediately affecting cell integrity, inducing lysis only as a later, secondary effect [42,71]. The complete (466-species) tree is available to download as an SI file (S1 Data); Fig 4 depicts a pruned (222-species) version of this tree in which up to 3 randomly chosen species from each represented genus are retained. For ease of reference, the species *Klebsiella pneumoniae*, *Serratia marcescens*, *Vibrio cholerae*, *Acinetobacter baylyi*, *Acinetobacter baumannii*, *Pseudomonas aeruginosa*, and *Burkholderia thailandensis* are protected from this pruning. Tree visualizations were produced using EMBL's interactive Tree of Life (iToL) viewer.

## Estimation of phylogenetic signal

Fig 4 shows that effectors with predicted lytic activity are present in high abundance across the phylum Proteobacteria. To examine whether this could be attributed to phylogeny alone (i.e., multiple species within a clade sharing an effector through common ancestry), we devised a simple similarity analysis to quantify phylogenetic signal (S8 Fig). First, we took the unpruned taxonomy tree (S1 Data) and divided it into a hierarchy of clades, grouping species occurring within ascending taxa (species in the same genus, the same family, and so on) up to the

"kingdom" taxon. For each clade $c$, we then computed species' within-clade similarity $S$ with respect to each effector $e$ in turn, using the following equation:

$$S_c^e = \frac{1}{N_c^2} \sum_{i=1}^{N_c} \sum_{j=1}^{N_c} \delta_{ij}(e); \quad \delta_{ij}(e) = \begin{cases} 1, \text{ if } i, j \text{ both have } e; \\ 1, \text{ if } i, j \text{ both lack } e; \\ 0, \text{ otherwise.} \end{cases}$$

Here, $N_c$ is the number of species in clade $c$, and $i, j$ are species' indices. Note that similarity $S_c^e = 1$ if all species in clade $c$ possess (or lack) at least 1 gene for effector $e$; otherwise $0.5 < S_c^e < 1$. We normalized these values by the similarity of the whole tree $S_{Tree}^e$, to give a similarity ratio $S_c^e / S_{Tree}^e$ for a given effector and taxon.

Similarity ratio values, plotted as a function of taxon ($c$), then provide a qualitative indicator of phylogenetic signal strength. S8A Fig provides an example of this: here, we show a cartoon taxonomy annotated with 3 imaginary effectors, X, Y, and Z. Effector X has a very strong phylogenetic signal, being conserved throughout the magenta clade, whereas effector Y is distributed randomly across the tree. Z is a rare effector, found at only 1 leaf of the tree. Plotting scaled similarity against taxon in S8B Fig, we see that X's similarity is approximately maximal for taxa 1 through 5, and then suddenly drops to the background level. In contrast, Y's similarity drops gradually. Note that individual clades containing rare effectors (e.g., Z) can have scaled similarity < global, because these clades are less self-similar than the tree is as a whole. Applying the same analyses to fast-lysing (S8C Fig) and slow-lysing effectors (S8D Fig), we generally see that similarity falls gradually rather than suddenly, indicating that phylogenetic signal is too weak to be the sole driver of effector abundance.

## Statistical analyses

Unless indicated otherwise, the number of biological replicates is 3 for each experiment. For assessing CFUs/mL, the decadic logarithm was taken. Average and standard deviation were calculated from logarithmic values. Normal distribution of values was checked using a D'Agostino-Pearson omnibus normality test. For comparative statistics, we used 2-way ANOVA ($\alpha$ = 0.05). Multiple comparisons were corrected for using a Tukey post hoc test. With the exception of Fig 3C (for which see "Osmoprotection assay"), all such calculations were performed in GraphPad Prism (v7.0).

## Supporting information

**S1 Fig. Agent-based modeling of T6SS firing and response.** (**A**) Diagram of 2D competition simulation showing initial ($t_{start}$; 100 cells) and final ($t_{end}$; approximately 10,000 cells) cell configurations. (**B**) Each simulation timestep, every T6SS+ cell fires $N_{firings,i,}$ i times with $N_{firings,i}$ drawn independently from a Poisson distribution with mean $k_{fire}$. (**C**) Cell's growth costs are computed from values of $N_{firings,i}$; costs are assumed to scale linearly with a cell's firing rate. (**D**) Cells respond to successful translocations with a steplike dose–response curve: once a cell's cumulative translocation count reaches threshold $N_{hits}$, that cell dies and becomes a "Victim"; see (E). Cells of the same genotype are immune to each other's effectors. (**E**) Cartoon of cell-based processes summarizing cell-based T6SS firing and response parameters; here, $k_{grow}$ is the maximum specific growth rate, and $k_{lys}$ the lysis rate post-T6SS intoxication. (**F**) Initial parameter sweep showing final attacker frequencies as a function of T6SS firing rate, for various weapon cost parameters $c$ (legend). Circles and bars denote means and standard deviations; 5 simulation replicates per case. Raw data are available at dx.doi.org/10.6084/m9.

figshare.11980491. T6SS, type VI secretion system.
(TIF)

**S2 Fig. Quantifying T6SS kill rate per unit interstrain boundary. (A)** Images of T6SS competition simulations, run as in Fig 1, highlighting a section of interstrain boundary between attacker (T6SS+, green) and susceptible (T6SS−, magenta) cell groups. **(B)** Diagrams of boundary cell classification (boundary cells shown with dashed outline) and of cell coordination number (5 neighbors of orange focal cell shown with dashed outline). **(C)** Boundary cell count traces (left axis) show number of T6+ cells in contact with nonkin cells (see B) as function of simulation time. Median cell coordination number (right axis, see B) plateaus to 5 ± 1 cells at confluency, after approximately 5 hours' growth. Black arrows correspond to the simulation snapshots shown in (A). **(D)** Absolute kill rates (blue traces, left axis) are measured by counting T6-dependent cell deaths per simulation step and then numerically computing the gradient of cumulative kill count trace (red traces, right axis). These traces are normalized by the number of boundary cells at each corresponding time point to give an NKR per unit interface (inset), which converges to a constant value in confluent colonies. **(E)** Normalized peak T6– kill rates taken at confluency (the maxima of the raw kill rates; vertical black lines in D), plotted against T6+ firing rate $k_{fire}$, for different weapon cost parameters $c$ (legend). Circles and bars denote means and standard deviations, respectively. Five simulation replicates per case. **(F)** Normalized peak T6– kill rates plotted against the ratio $k_{fire} / N_{hits}$, for different lysis rates $k_{lysis}$. Symbols denote the value of the lethal hit threshold $N_{hits}$ used in each simulation (legend). For each of the 6 resulting simulation groups, we found that increasing $N_{hits}$ was equivalent to proportionally reducing $k_{fire}$, such that plotting peak kill rates against the ratio $k_{fire} / N_{hits}$ yielded the same curve for each $k_{lysis}$ value. Black lines correspond to Monod curves, fitted for each $N_{hits}$ value, as a test of their similarity (solid, $k_{lysis} = 1.6$ h$^{-1}$; dashed, $k_{lysis} = 0.8$ h$^{-1}$). Ten simulation replicates per case. **(G, H)** Magnified sections of simulated communities show occupation of interstrain boundary by lysing cells (see legend) at increasing T6SS firing rates. As in Fig 1, "slow lysis" and "rapid lysis" cases correspond to $k_{lysis} = 0.8, 8.0$ h$^{-1}$, respectively. **(I)** Boundary saturation, computed as the fraction of interstrain boundaries occupied by lysing cells, is shown as a function of simulation time, for each firing and lysis rate in (G, H). Arrows indicate time points depicted in snapshots. Raw data: dx.doi.org/10.6084/m9.figshare.11980491. NKR, normalized kill rate; T6SS, type VI secretion system.
(TIF)

**S3 Fig. Automated image analysis for microfluidic experiments. (A)** Diagram of microfluidic chip (left), showing inlets, outlets, flow, and observation channels; zoomed section (right) shows observation channels loaded with *A. baylyi* and T6SS− *E. coli*. Diagram not to scale. Flow diagram **(B)** summarizing workflow for image postprocessing and segmentation in Fiji. The graph shown can be found in S5C Fig. **(C, D)** Examples for image preprocessing (contrast enhancement, blurring, and background subtraction) are shown in the adjusted composite image. GFP and mRuby3 fluorescence signals were used for segmentations (turquois outlines). Examples are provided for low **(C)** and high **(D)** chamber occupancies. Scale bar: 2 μm. GFP, green fluorescent protein; mRuby3, monomeric red fluorescent protein; T6SS, type VI secretion system.
(TIF)

**S4 Fig. Fast-lysing effector Tae1 displays higher prey cell killing rate compared with slow-lysing Tse2.** Fast-lysing **(A)** and slow-lysing **(B)** single-effector *A. baylyi* attacker strains (*vipA-sfGFP*, green) were coincubated with *E. coli* (unlabeled) victims in the presence of 2 μg ml$^{-1}$ PI (PI dead stain, magenta), within microfluidic channels. Two additional examples are

displayed here, analogous to those shown in Figs 2A and 2B. The percentage reduction in *E. coli* channel occupancy over 8 hours is indicated, with t = 0 corresponding to the point at which the chamber becomes confluent. To measure the rate of victim cell death, each time-lapse series was split into 1-hour segments (consisting of 12 frames, 5-minute acquisition frame rate) from the moment of confluency, and the number of new PI foci appearing after each hour were counted. Examples of 3 consecutive time points are provided in the far-right column; yellow triangles highlight cell death event. Then, the number of cell death events per hour was normalized to the contact perimeter between *E. coli* and *A. baylyi* (based on sfGFP signal at first frame of every 1-hour segment). From this, the victim cell kill rate over time **(C)** and thence the average victim cell killing rate per hour **(D)** were determined. This analysis was carried out for 10 separate microchannels for both attacker strains. Scale bars: 2 μm. Raw data are available at dx.doi.org/10.6084/m9.figshare.11980491. PI, propidium iodide; sfGFP, super-folding green fluorescent protein; Tae1, type VI amidase effector 1; Tse2, type VI effector 2; VipA, ClpV-interacting protein A.
(TIF)

**S5 Fig. Microfluidic chambers: Time series, occupancy traces, and additional simulations.** Fluorescence time-lapse series showing microfluidic competition experiments, between *E. coli* expressing cytosolic mRuby3 (magenta) and *A. baylyi* expressing *vipA-sfGFP* (green) armed with different T6SS effectors. **(A)** *A. baylyi* secreting Tae1. **(B)** *A. baylyi* secreting Tse2. **(C)** *A. baylyi* Parental strain. **(D)** Nonsecreting *A. baylyi* T6SS− mutant (*Δhcp*). Below each time series, the corresponding strain area occupancy plot, computed using the procedure described in S3 Fig, is shown. (A, B) are reproduced from Fig 2 but show full channel overview. Additional time-lapse series are shown in S3 Movie (A, B) and S4 Movie (C, D). Scale bar: 2 μm. **(E)** Color map representing final (= 16 hours) *E. coli* occupancy in chamber simulations, performed as in Fig 2, for different $k_{fire}$ and $k_{lysis}$ parameter values. Four example strain occupancy plots (1–4) are marked and shown below. Legends as in (A–D); dashed lines and bars denote occupancy means and standard deviations, respectively. Five simulation replicates per case. Raw data are available at dx.doi.org/10.6084/m9.figshare.11980491. mRuby3, monomeric red fluorescent protein; PI, propidium iodide; sfGFP, super-folding green fluorescent protein; Tae1, type VI amidase effector 1; Tse2, type VI effector 2; VipA, ClpV-interacting protein A.
(TIF)

**S6 Fig. Fluorescence images and cell recovery for agarose competition experiments.** Surface competition assays comparing performance of T6SS+ attacker *A. baylyi* (*vipA-sfGFP*, green) armed with lytic (Tae1) and nonlytic (Tse2) effectors, competing with susceptible *E. coli* (*mRuby3*, magenta). Representative phase-contrast and fluorescence micrographs show pre- and post-3-hour, 37°C coincubation distributions of *A. baylyi* and *E. coli* cells, for mixtures of parental *A. baylyi*, Tae1, and Tse2 single-effector strains and T6SS− *Δhcp* mutant. Fluorescence signal for each channel was blurred and background subtracted prior to Otsu segmentation (rightmost column). Based on this, the respective area occupancies of *A. baylyi* and *E. coli* within the community were quantified; these data are plotted in Fig 2G. White pixels indicate overlapping signals for both attacker and susceptible strain. Scale bar = 50 μm; inserts depict a 50 × 50 μm field of view. Hcp, hemolysin-coregulated protein; mRuby3, monomeric red fluorescent protein PI, propidium iodide; sfGFP, super-folding green fluorescent protein; Tae1, type VI amidase effector 1; Tse2, type VI effector 2; VipA, ClpV-interacting protein A.
(TIF)

**S7 Fig. EDTA induces lysis in wall-compromised *E. coli*, negating OP.** Micrographs show *E. coli* cells following 3-minute (top row) and 33-minute incubation (bottom row) on agarose

pads, following different pretreatment conditions (column labels). OP prevents cell lysis from cell wall damage sustained from Amp (middle column); addition of 20 mM EDTA to pad media (LB) negates osmoprotection, resulting in mass lysis (right column). Scale bar: 2 μm. Amp, ampicillin; LB, lysogeny broth; OP, osmoprotectant.
(TIF)

**S8 Fig. Phylogenetic signal in T6SS effector distribution among Proteobacteria. (A)** Example taxonomy annotated with 3 effectors: "X," common only to magenta clade, corresponding to strong phylogenetic signal; "Y," randomly distributed across tree, weak phylogenetic signal; "Z," rare effector found only in only 1 species. **(B)** Plot of similarity ratio (within-clade similarity, $S_c^e/S_{Tree}^e$, see Materials and methods) versus taxon for the 2 effectors shown in (A). Individual points mark similarity ratios within a given clade; lines correspond to mean similarity ratio. **(C)** Plot of similarity ratios, analogous to (B), for the fast-lysing effectors shown in Fig 4. Values are scaled such that all effectors share the same maximum ("Max." $= 1/S_{Tree}^e$) and converge to the same similarity ratio ("Global" $= 1$). Horizontal axis labels denote the following taxa: 1 = subspecies, 2 = species, 3 = subgenus, 4 = genus, 5 = family, 6 = order, 7 = class, 8 = phylum, 9 = kingdom (full tree). **(D)** Plot of similarity ratios, analogous to that shown in (C), for the slow-lysing effectors shown in Fig 4. Raw data are available from [dx.doi.org/10.6084/m9.figshare.11980491](http://dx.doi.org/10.6084/m9.figshare.11980491). T6SS, type VI secretion system.
(TIF)

**S1 Movie. Simulated T6SS competitions at different firing rates.** Representative agent-based model simulations show communities growing from a randomly scattered 1:1 inoculum of T6SS+ attacker cells (green) and T6SS− susceptible cells (magenta). Here, monolayers of 3D rod-shaped cells grow exponentially through elongation, dividing after doubling their initial volume, and pushing on neighboring cells during expansion. T6SS+ cells attack T6SS− cells with discrete, randomly oriented needles, fired at average rate $k_{fire}$; intoxicated T6SS− cells (black) cannot divide, and lyse after a time delay $1/k_{lysis}$. From left to right, the T6SS firing rate is increased (0, 50, and 250 firings cell$^{-1}$ h$^{-1}$, respectively). Simulation parameters: $N_{hits} = 1$, $c = 0.001$, $k_{lysis} = 0.8, 8.0$ h$^{-1}$. Screen measures approximately $700 \times 350$ μm; frame rate 5 frames per second. T6SS, type VI secretion system.
(AVI)

**S2 Movie. Simulations of microfluidic chamber competitions.** Representative agent-based model simulations of T6SS+ attacker cells (green) competing with T6SS− susceptible cells (magenta) in $100 \times 10$ μm microfluidic chambers, under conditions of fast (top, $k_{lysis} = 8.0$ h$^{-1}$) and slow (bottom, $k_{lysis} = 0.8$ h$^{-1}$) victim lysis. Lysing T6SS− cells are shown in black; initial cell positions are identical. Simulation parameters: $N_{hits} = 1$, $c = 0.001$, $k_{fire} = 50$ firings cell$^{-1}$ h$^{-1}$. Screen measures approximately $115 \times 58$ μm; frame rate 5 frames per second. T6SS, type VI secretion system.
(MP4)

**S3 Movie. Microfluidic competition assay comparing *E. coli* survival between lytic and nonlytic *A. baylyi* single-effector strains.** T6SS+ *A. baylyi* (*vipA-sfGFP*, green) were competed with T6SS− susceptible *E. coli* (*mRuby3*, magenta) cells in 1-μm-deep microfluidic channels. The fast-lysing *tae1* single-effector aggressor strain is displayed on the left; slow-lysing *tse2* single-effector aggressor strain is displayed on the right. Five representative time-lapse series acquired with a rate of 5 minutes per frame are shown. sfGFP and mRuby3 fluorescence channels are displayed individually as grayscale images in addition to a merge of phase-contrast and both fluorescence channels. Scale bar: 2 μm. The video plays at a rate of 12 frames per second. mRuby3, monomeric red fluorescent protein; sfGFP, super-folding green fluorescent

protein; T6SS, type VI secretion system; Tae1, type VI amidase effector 1; Tse2, type VI effector 2; VipA, ClpV-interacting protein A.
(MP4)

**S4 Movie. Microfluidic competition assay in presence or absence of T6SS-mediated prey cell killing.** Parental (left) and isogenic T6SS– (Δ*hcp*, right) *A. baylyi* (*vipA-sfGFP*, green) strains were competed with T6SS– susceptible *E. coli* (*mRuby3*, magenta) cells in 1-μm-deep microfluidic channels. A representative time-lapse series acquired with a rate of 5 minutes per frame is shown. sfGFP and mRuby3 fluorescence channels are displayed individually as grayscale images in addition to a merge of phase-contrast and both fluorescence channels. Scale bar: 2 μm. The video plays at a rate of 12 frames per second. Hcp, hemolysin-coregulated protein; mRuby3, monomeric red fluorescent protein; sfGFP, super-folding green fluorescent protein; Tae1, type VI amidase effector 1; Tse2, type VI effector 2; VipA, ClpV-interacting protein A.
(MP4)

**S5 Movie. Microfluidic competition assay in presence of propidium iodide.** Fast-lysing (Tae1, left) and slow-lysing (Tse2, right) single-effector attacker strains were coincubated with *E. coli* victim strain, in the presence of 2 μg ml$^{-1}$ propidium iodide dead stain. Cell were imaged for 15 hours at a 5-minute acquisition frame interval. The video plays at a rate of 12 frames per second. Tae1, type VI amidase effector 1; Tse2, type VI effector 2.
(MP4)

**S6 Movie. OP conditions prevent target cell lysis.** T6SS+ *A. baylyi* attacker strain (*vipA-sfGFP*, green) armed with Tae1 was competed with T6SS– susceptible *E. coli* cells on LB 1% agarose pads containing 2 μg/mL propidium iodide (magenta). Agarose pads were left untreated (–OP, left) or supplemented with 0.4 M sucrose, 8 mM MgSO$_4$ (+OP, right). Ten representative 45-minute time-lapse series acquired with a rate of 30 seconds per frame are shown. Field of view is 10 × 10 μm and shows a merge of phase-contrast and 2 fluorescent channels: VipA-sfGFP and propidium iodide. Scale bar: 2 μm. The video plays at a rate of 12 frames per second. LB, lysogeny broth; OP, osmoprotectant; sfGFP, super-folding green fluorescent protein; T6SS, type VI secretion system; Tae1, type VI amidase effector 1; Tse2, type VI effector 2; VipA, ClpV-interacting protein A.
(MP4)

**S7 Movie. Microfluidic competition experiment in presence of OP conditions.** T6SS+ *A. baylyi* attacker strain (*vipA-sfGFP*, green) was competed with T6SS– susceptible *E. coli* (*mRuby3*, magenta) cells in presence of OP in 1-μm-deep microfluidic channels. Five representative time-lapse series acquired with a rate of 5 minutes per frame are shown. sfGFP and mRuby3 fluorescence channels are displayed individually as grayscale images in addition to a merge of phase-contrast and both fluorescence channels. Scale bar: 2 μm. The video plays at a rate of 12 frames per second. OP, osmoprotectant; T6SS, type VI secretion system; VipA, ClpV-interacting protein A; sfGFP, super-folding green fluorescent protein; mRuby3, monomeric red fluorescent protein.
(MP4)

**S1 Table. Model parameters used in this study.**
(XLSX)

**S2 Table. Model variables used in this study.**
(XLSX)

**S3 Table. Bacterial strains used in this study.**
(XLSX)

**S1 Data. Unpruned version of Fig 4.**
(SVG)

## Acknowledgments

We are grateful to Elisa Granato, Oliver Meacock, Connor Sharp, Daniel Unterweger, Jonas Schluter, and Wook Kim for their insight and comments and to Matteo Sangermani for his assistance with setting up microfluidics in the Basler Lab. We would also like to thank the Biozentrum's imaging core facility for their advice on image analyses.

## Author Contributions

**Conceptualization:** William P. J. Smith, Andrea Vettiger, Laurie E. Comstock, Marek Basler, Kevin R. Foster.

**Funding acquisition:** Laurie E. Comstock, Marek Basler, Kevin R. Foster.

**Investigation:** William P. J. Smith, Andrea Vettiger, Marek Basler, Kevin R. Foster.

**Methodology:** William P. J. Smith, Andrea Vettiger, Julius Winter, Till Ryser, Marek Basler.

**Resources:** Laurie E. Comstock, Marek Basler, Kevin R. Foster.

**Software:** William P. J. Smith.

**Supervision:** Laurie E. Comstock, Marek Basler, Kevin R. Foster.

**Writing – original draft:** William P. J. Smith, Andrea Vettiger.

**Writing – review & editing:** William P. J. Smith, Andrea Vettiger, Laurie E. Comstock, Marek Basler, Kevin R. Foster.

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
