## [Editor Report · Decision Letter 0]

5 Dec 2019

Dear Kevin, 

Thank you for submitting your manuscript entitled "The Evolution of the Type VI Secretion System as a Disintegration Weapon" for consideration as a Research Article by PLOS Biology.

I've assessed your manuscript and consult with an academic editor with relevant expertise and I am writing to let you know that we would like to send your submission out for external peer review.

Please re-submit your manuscript within two working days, i.e. by Dec 07 2019 11:59PM.

Kind regards,

Lauren

Lauren A Richardson, Ph.D

Senior Editor

PLOS Biology

---

## [Decision Letter · Decision Letter 1]

14 Jan 2020

Dear Dr Foster,

Thank you very much for submitting your manuscript "The Evolution of the Type VI Secretion System as a Disintegration Weapon" for consideration as a Research Article at PLOS Biology. Your manuscript has been evaluated by the PLOS Biology editors, an Academic Editor with relevant expertise, and by several independent reviewers.

As you will read, the reviewers appreciated many aspects of your study. However, they also raise some concerns that will need to be addressed in a revision. Two of the reviewers suggest, but do not require, bolstering your conclusions with studies of additional toxins. We agree that this would improve the study but will leave it to your discretion whether to include further data.

In light of the reviews (below), we will not be able to accept the current version of the manuscript, but we would welcome re-submission of a much-revised version that takes into account the reviewers' comments. We cannot make any decision about publication until we have seen the revised manuscript and your response to the reviewers' comments. Your revised manuscript is also likely to be sent for further evaluation by the reviewers.

We expect to receive your revised manuscript within 2 months. 

**IMPORTANT - SUBMITTING YOUR REVISION**

*NOTE: In your point by point response to to the reviewers, please provide the full context of each review. Do not selectively quote paragraphs or sentences to reply to. The entire set of reviewer comments should be present in full and each specific point should be responded to individually, point by point.

*Re-submission Checklist*

*Published Peer Review*

*PLOS Data Policy*

*Blot and Gel Data Policy*

Sincerely,

Lauren A Richardson, Ph.D

Senior Editor

PLOS Biology

REVIEWS:

Reviewer #1: Robert M. Cooper, signed review

This manuscript describes a significant and novel insight into the ecological significance and strategy of bacterial combat using the Type VI Secretion System (T6SS). In particular, injected toxins that cause rapid, catastrophic lysis of the victim cell are more effective weapons than toxins that result in a less violent demise. This is because if the victims remain intact, their remains form a "corpse layer" that shields the rest of their colony from the T6SS attackers. The ecological function of the T6SS remains relatively unexplored, and this paper represents a significant advance on understanding its function in microbial communities. The manuscript supports this interpretation using an elegant series of experiments, simulations, and analysis. These include agent-based simulations, bulk competitions on agar plates, microscopic observations in microfluidic chips, and a phylogenetic analysis of T6SS toxins. A set of experiments using osmoprotectants to prevent lysis by an otherwise lytic toxin was particularly clever, as it demonstrated that the combat advantage of lytic toxins is directly due to lysis.

One set of experiments that would significantly bolster the authors' interpretation would be to repeat the microfluidic experiments with A. baylyi strains containing only its other lytic T6SS effector Tle1 and its other nonlytic effector Tse1. This would experimentally demonstrate generalizability of the effect and result in a stronger paper, particularly if these effectors have different killing rates from those already used, thus sampling more than just two points of the parameter space. However, if these strains are not on hand or the experiments would require a significant amount of time, the paper would still stand without them.

Significant points:

1. Line 456: Killing dependence on interfacial saturation is given as k_kill=f_interface * k_fire * p_hit. However, f_interface is defined elsewhere as the fraction of the interface covered by dead cells, and increasing f_interface should decrease k_kill by providing a barrier layer. So should the dependence actually be (1-f_interface)?

2. Figure 2a,b using death stains to show a dead corpse layer for the non-lytic toxin are very important to the interpretation, but 2a seems to have significant cell death in the interior of the E. coli patch. Is this because A. baylyi have pushed inwards past dead E. coli, creating an overlap between the predator (green) and dead prey (magenta) signals? If so, it would be helpful to show the green channel by itself, as done for the magenta channel. If not, that would imply some degree of nonlocal killing, which would present a problem for the corpse barrier interpretation. If the death stain is spreading into the interior because DNA released by lysis is migrating inward, including a movie for these death stain experiments would be useful (again showing each channel separately as well as the merge).

3. I was confused by the description of normalized peak kill rates. The caption for Figure S2 (line 48) says "the maxima of these traces… are normalised by the number of boundary cells at each corresponding time point", but the inset to Fig. S2d shows time course data, so it must be more than just the maximum rates that were normalized. Were all time points of a given trace normalized by the number of boundary cells at the time of the maximum, or was each time point normalized by its own number of boundary cells? This could also be clarified in Methods (line 449) by specifying in the equation at what time point N_boundary is calculated.

Minor points:

1. The discussion of optimal T6SS firing rates due to a tradeoff between energy costs vs killing victims is specific to unregulated T6SS firing, unlike in e.g. Pseudomonas aeruginosa, which only counter-attacks after being attacked first and directionally concentrates its fire on the aggressor, thus conserving resources. It would be good to add this caveat to the interpretation.

2. Coordination number is referenced but never defined.

3. Line 237, Figure 3c caption: "toxin translocation rates" would more accurately be "kill rates", as this is what was directly measured and does not rely on an assumption about the number of required hits.

Format, etc:

1. Line 162: extra "strain"

2. Line 514: missing verb (occupancy was calculated)

3. Line 538: missing "and"

4. Data and code availability: still has placeholder XXX's

5. I was unable to open Movie S1 using Quicktime Player, although this opened the other movies, and I was able to open Movie S1 using VLC. The authors may wish to use a more widely compatible codec for Movie S1.

--------------

Reviewer #2: 

This is an excellent paper examining the role of toxin type in the evolutionary advantage of the T6SS. While a great deal is known about the mechanism and regulation of the T6SS, little is known about the factors driving its evolution. This paper provides a really interesting insight into this evolution, showing that lytic toxins delivered by the T6SS make it a more potent weapon as dead but non-lysed cells form a barrier preventing further attack of enemy strains. The authors then go on to show that lytic toxins are common across the T6SS wielding bacteria. 

The methods used by the authors in their simulations and experiments are state of the art, providing good mechanistic insight into how the "corpse barrier effect" operates. The quantitative agreement between the simulation models and experiments is particularly striking, providing further evidence that the corpse barrier effect is the driver of their experimental results. The manuscript is also written very clearly and the discussion of the results well balanced. I think this will make a great addition to the literature and plos biology would make a great home for it. I have only two suggestions to improve the manuscript.

Firstly, while the authors experiments are elegant and careful in design, it is important to note that only one lytic and one non-lytic toxin were compared. This is understandable given the detailed nature of the authors experiments, and I am not suggesting that they carry out further experiments with additional toxins as this would be beyond the scope of the manuscript, and they have supported their predictions using comparative phylogenetics. However, I think it should be somewhat caveated that their results are for a single pair of toxins in the manuscript. This in itself provides rationale for their phylogenetic analysis, so should be easy to integrate into the manuscript.

Secondly, in their phylogenetic analysis the authors simply report proportions of species and effectors that show lysis. While these proportions obviously support the authors' hypothesis, there are issues with trusting such proportions in phylogenetic studies. For example, if a very large single clade of closely related species has an over-representation of lytic toxins, then this could lead to an artificially inflated estimate of the tendency for lytic toxins to evolve. While it does not look like this is the case from the authors' phylogenetic plots, it would still be best to control for phylogeny in making these estimates. The state of the art approach here would be to fit a Bayesian phylogenetic mixed model controlling for phylogeny as a random effect (can be implemented in R using MCMCglmm), and take the estimate of the proportion of species expected to have lytic toxins from this model. This would allow for an estimate of the evolutionary tendancy for T6SS toxins to be lytic while removing any confounding because of the phylogenetic distribution of the species sampled.

--------------

Reviewer #3: Martin Ackermann, signed review

This manuscript by William Smith and colleagues focuses on the cellular dynamics of contact-dependent killing in bacterial populations. Antagonistic interactions are common between microbial strains, and a wide-spread and prominent mechanism is based on the translocation of toxic proteins from one cell to another via a molecular machinery known as type VI secretion system. While the molecular mechanisms of type VI secretion are well known and widely studied, much less is known about the functional consequences and benefits that arise from killing. Generally, the type VI system is understood as a mechanism to eliminate competitors, but how this works exactly is not well understood. We do not know how efficiently competing cells are killed by this system, and how the efficiency depends on whether cells are planktonic or surface-attached. Addressing this knowledge gap is important for at least two reasons. First, it provides a functional and ecological context to the vibrant field of research that focuses on molecular mechanisms of type VI secretion. Second, type VI secretion is receiving increasing attention in an applied context, primarily in the context of microbial interactions in host-associated microbiomes (including the gut microbiome); understanding the dynamics of type VI-mediated killing is fundamental to any such applications. 

Willliam Smith and colleagues combine computer simulations, single-cell analysis and comparative genomics to test one main hypothesis about the dynamics of type VI-mediated killing: the hypothesis is that, in situations where microbes grow on surfaces, type VI-mediated killing is much more efficient if the target cell is lysed than if the target cell is only killed but remains physically intact. The rationale for this hypothesis is as follows: if target cells remain intact, strains that carry a type VI-secretion system become rapidly surrounded by a layer of dead but physically intact target cells, preventing the type VI-secretion carrying strain to expand further through microbial assemblies. The authors find support for this hypothesis from their simulation and from the single-cell experiments. In addition, the analysis of bacterial genomes indicates that a large fraction of bacteria with type VI secretion systems translocate proteins that lyse the target cell, rather than just killing it. 

In my opinion, this manuscript makes an important and original contribution to our understanding of microbial interactions, a contribution that has implications for both fundamental research as well as applied purposes. Also, the combination of modeling, genetic manipulation, quantitative single-cell analysis and comparative genomics is unique and raises the bar in the field of microbial ecology and evolution. 

I have a few concerns and questions that I think should be addressed. 

A first main point (which is also mentioned by the authors on lines 207ff) is that the main experimental test has a potential caveat, namely the possibility of different killing rates mediated by the lytic and non-lytic effectors. The authors do address this issue by performing experiments with an osmo-protectant. I agree that these experiments offer additional support for their hypothesis, but think that a more detailed analysis of the single-cell experiments would potentially allow for a more direct test. Can the authors directly establish that, for the non-lytic effector, producer cells (i.e., type VI-carrying cells) and live target cells become separated by a layer of dead target cells? And can they show that this effect decreases the death rate of live target cells that are separated from the producer cells by this layer? I realize that this quantification might be challenging but think that it would provide strong and direct support for the main hypothesis that the authors put forward. Related to this point, visual inspection of the lower panel of Fig. 2b gave me the impression that target cells can be killed even if they are not in direct contact with producer cells but rather several cell layers away from them. The quantification proposed above would potentially resolve this issue and provide more quantitative understanding of how spatial positioning of producers and dead and live target cells determines killing rates. 

Smaller comments

In the section about the simulation, the authors often use the term "measurements" to refer to data extracted from the simulation. I think this could be potentially confusing for readers who might misunderstand this as referring to experimentally generated data. 

Fig. 1b, d: the point that fast lysis reduces the layer of dead but intact cells does not become really clear to me from these panels (the yellow arrow don't help that much). I think a quantification of the fraction of the interstrain boundary that is occupied by dead but intact cells would be more informative. 

Line 115: the term "interfacial saturation" is maybe not self-explanatory. Consider using a simpler term?

Fig. 2e, f: what experimental data was used to generate these plots? Is this data from several biological replicates? If yes, how many? What is the variation between measurements from different replicates?

Fig. 2e, f: I think it would be important to test the hypothesis that target cells maintain higher proportions in the case of non-lytic effector proteins directly with a statistical test. 

Lines 218 and 220: the authors write that they find "equal amounts of … protein" and "equal numbers of cell lysis". I think it is fundamentally not possible to establish that two experimentally measured values are "equal"; we can only report that we find no evidence that they are significantly different - but the absence of significance can have many reasons, for example a high error variance. I suggest that the authors instead make a statement about how small (or large) the difference between the two measurements is, for example by giving a confidence interval for the difference between the two measurements. 

Line 273: the numbers are not clear to me (83.2% are predicted to cause lysis and 84.8% carry at least one lytic effector - why are then not 84.8% predicted to cause lysis?)

In the section about the genomic analysis, the authors suggest that DNases and pore-forming toxins do not lead to cell lysis (lines 276 and 290). Why I see that this is plausible, I would be interested in seeing more specific support for this statement - is this point supported by empirical evidence? I don't know much about the lytic effect of different type VI effectors, but for example in antibiotics the link between mode of action and cell lysis is not always obvious.

---

## [Decision Letter · Decision Letter 2]

1 Apr 2020

Dear Dr Foster,

Thank you for submitting your revised Research Article entitled "The Evolution of the Type VI Secretion System as a Disintegration Weapon" for publication in PLOS Biology. I have now obtained advice from original reviewer 1 and the academic editor. 

Based on their assessments and advice, we will probably accept this manuscript for publication, assuming that you will modify the manuscript to address the remaining points raised by reviewer 1. Please also make sure to address the data and other policy-related requests noted at the end of this email.

We expect to receive your revised manuscript within two weeks. Your revisions should address the specific points made by reviewer 1. In addition to the remaining revisions and before we will be able to formally accept your manuscript and consider it "in press", we also need to ensure that your article conforms to our guidelines. A member of our team will be in touch shortly with a set of requests. As we can't proceed until these requirements are met, your swift response will help prevent delays to publication.

*Copyediting*

*Published Peer Review History*

*Early Version*

*Submitting Your Revision*

Sincerely,

Di Jiang, PhD

Associate Editor

PLOS Biology

DATA POLICY:

Regardless of the method selected, please ensure that you provide the individual numerical values that underlie the summary data displayed in the following figure panels as they are essential for readers to assess your analysis and to reproduce it: Figures 1gh, 2efgh, 3bcef, S1bcdf, S2cdefj, S3b, S4abcde, S5cd, S6bcd, unless they are included in FigShare file repository (dx.doi.org/10.6084/m9.figshare.11980491); if so, please provide us with an editor/reviewer access key/token so that we can check the data before we can accept the paper. NOTE: the numerical data provided should include all replicates AND the way in which the plotted mean and errors were derived (it should not present only the mean/average values).

Reviewer remarks:

Reviewer #1 (Robert M Cooper, signed review): The updates in this revised version address nearly all of my points, and I look forward to adding the finished paper to my library. The new Movie S5 is a useful addition that shows killing is concentrated at the species boundary.

My only remaining suggestion is to clarify the legend for Figure S2d, specifically regarding the inset. The portion I'm referring to reads:

"The maxima of these traces (vertical black lines) are normalised by the number of boundary cells at each corresponding timepoint to give a kill rate per unit interface (inset), which converges to a constant value in confluent colonies."

If I'm interpreting this correctly (as per the updated Methods at line 469), "maxima" should be removed when referring to the inset. The inset appears to show just the Normalized Killing Rate (NKR) with no maxima taken. The max doesn't seem to come in until the next panel S2e with normalized PEAK killing rate.

So perhaps replace that sentence in the legend with something like: "These traces are normalised by the number of boundary cells at each corresponding timepoint to give a kill rate per unit interface (inset), which converges to a constant value in confluent colonies" and change the legend for E to "Normalized peak T6- kill rates taken at confluence (the maxima of the raw kill rates, vertical black lines in D), plotted against...

---

## [Editor Report · Decision Letter 3]

30 Apr 2020

Dear Dr Foster,

On behalf of my colleagues and the Academic Editor, Victor Sourjik, I am pleased to inform you that we will be delighted to publish your Research Article in PLOS Biology. 

Early Version

PRESS 

Kind regards,

Alice Musson

Publishing Editor, 

PLOS Biology

on behalf of

Di Jiang,

Associate Editor

PLOS Biology